# Tractable Transformers for Flexible Conditional Generation

Anji Liu [*1 2]   Xuejie Liu [*3 4]   Dayuan Zhao [5]   Mathias Niepert [2]   Yitao Liang [3]   Guy Van den Broeck [1]

## Abstract

Non-autoregressive (NAR) generative models are valuable because they can handle diverse conditional generation tasks in a more principled way than their autoregressive (AR) counterparts, which are constrained by sequential dependency requirements. Recent advancements in NAR models, such as diffusion language models, have demonstrated superior performance in unconditional generation compared to AR models (e.g., GPTs) of similar sizes. However, such improvements do not always lead to improved conditional generation performance. We show that a key reason for this gap is the difficulty in *generalizing to conditional probability queries (i.e., the set of unknown variables) unseen during training*. As a result, strong unconditional generation performance does not guarantee high-quality conditional generation. This paper proposes Tractable Transformers (Tracformer), a Transformer-based generative model that is more robust to different conditional generation tasks. Unlike existing models that rely solely on global contextual features derived from full inputs, Tracformers incorporate a sparse Transformer encoder to capture both local and global contextual information. This information is routed through a decoder for conditional generation. Empirical results demonstrate that Tracformers achieve state-of-the-art conditional generation performance on text modeling compared to recent diffusion and AR model baselines. Code is available at https://github.com/liuanji/Tracformer.

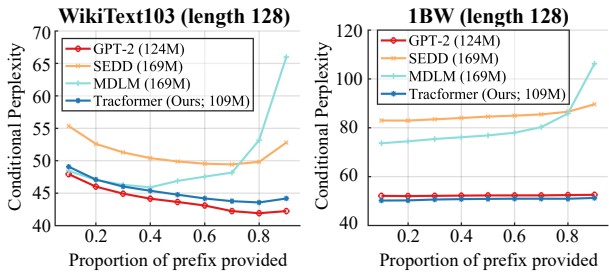

Figure 1: Zero-shot conditional perplexity (↓) of AR (GPT-2 (Radford et al., 2019)) and NAR models (SEDD (Lou et al., 2023), MDLM (Sahoo et al., 2024)) models conditioned on different proportions of prefix given sequences of length 128. Existing NAR models perform worse than GPT-2 despite having comparable unconditional perplexity on sequences of length 1024. Their performance gets worse when provided with more context.

## 1. Introduction

Generative AI has emerged as a transformative paradigm for solving machine learning tasks. Its core strength lies in the ability of modern deep generative models to learn complex and high-dimensional data distributions. Autoregressive (AR) models such as GPTs (Brown et al., 2020) are among the most well-developed generative models, demonstrating exceptional performance in modeling discrete data such as language (Dubey et al., 2024; Reid et al., 2024) and protein sequences (Shin et al., 2021; Trinquier et al., 2021). However, despite their expressiveness and scalability, AR models are not best suited for many conditional generation tasks such as DNA imputation and protein sequence infilling (Tanaka et al., 2022; Hawkins-Hooker et al., 2023; Alamdari et al., 2023) due to their inherent (autoregressive) sequential dependencies (Kaddour et al., 2023).

In contrast, non-autoregressive (NAR) generative models are inherently more flexible, as they can condition on arbitrary contexts. This flexibility makes them particularly well-suited for conditional generation tasks, such as code editing (Liu et al., 2024) and DNA imputation (Stark et al., 2024; Li et al., 2024; DaSilva et al., 2024). Recent advances in discrete diffusion (Lou et al., 2023; Shi et al., 2024) and semi-autoregressive models (Chen et al., 2024) have further improved the performance of NAR models across various domains. Notably, SoTA diffusion language models outperform GPT-2 in terms of unconditional perplexity on several benchmarks (Sahoo et al., 2024).

---

[*]Equal contribution  [1]Department of Computer Science, University of California, Los Angeles [2]Institute for Artificial Intelligence, University of Stuttgart [3]Institute for Artificial Intelligence, Peking University [4]School of Intelligence Science and Technology, Peking University [5]Yuanpei College, Peking University. Correspondence to: Anji Liu <liuanji@cs.ucla.edu>, Yitao Liang <yitaol@pku.edu.cn>, Guy Van den Broeck <guyvdb@cs.ucla.edu>.

*Proceedings of the 42nd International Conference on Machine Learning*, Vancouver, Canada. PMLR 267, 2025. Copyright 2025 by the author(s).

However, we observe a significant performance drop when using SoTA diffusion language models for conditional generation tasks. As shown in Figure 1, we evaluate the conditional perplexity (i.e., the perplexity of the corresponding conditional likelihood) of two SoTA diffusion language models, SEDD (Lou et al., 2023) and MDLM (Sahoo et al., 2024) by providing different fractions of the prefix of length-128 sequences from WikiText103 (Merity et al., 2022) and 1BW (Chelba et al., 2013). Despite both models having comparable or better unconditional perplexity than GPT-2 on length-1024 sequences, their conditional perplexity is significantly worse. Moreover, their performance degrades further as the length of the provided prefix context increases.

We attribute this performance discrepancy to the inability of existing NAR models to generalize effectively to conditional queries unseen during training. To address this issue, we propose **Tractable Transformers (Tracformers)**, an NAR generation model designed to handle diverse conditional generation tasks more robustly. A key insight is to learn local features that enhance generalization across different queries. Specifically, unlike existing models that rely solely on global features from all input tokens, Tracformers use a novel sparse encoder to learn features at multiple context levels, which are then processed by a decoder for conditional generation. As shown in Figure 1, Tracformer achieves better conditional perplexity than other NAR models.

Empirical results on text generation show that Tracformer achieves consistently better conditional generation performance compared to existing architectures such as BERT (Devlin et al., 2018) and BART (Lewis et al., 2020). Further, Tracformer beats state-of-the-art (SoTA) diffusion language models on zero-shot conditional generation tasks, which are conditional variants of the zero-shot perplexity tasks used to evaluate GPT-2 (Radford et al., 2019). In addition to proposing an NAR model architecture, we emphasize the importance of directly evaluating the conditional generation performance of NAR models.

In summary, the contributions of this paper are two-fold: (i) We identified the problem that existing NAR models suffer from severe performance degradation in conditional generation, despite having strong unconditional generation performance; (ii) We propose Tracformer, a novel Transformer-based architecture specially designed to improve conditional generation and generalize to conditional queries unseen during training.

## 2. Background

In this section, we first introduce key concepts of sequence modeling and the distinctions between autoregressive and non-autoregressive approaches (Sec. 2.1). We then describe the Transformer architecture, which serves as the backbone

of many modern sequence modeling frameworks, highlighting its key components and their roles in both AR and NAR paradigms (Sec. 2.2). We use uppercase letters (e.g., $X_t$) to represent random variables and lowercase letters (e.g., $x_t$) for their assignments.

### 2.1. Sequence Modeling

Given a sequence of $T$ categorical variables $\mathbf{X} := \{X_t\}_{t=1}^T$, sequence modeling aims to capture their joint distribution. Autoregressive modeling achieves this by factorizing the joint probability of $\boldsymbol{x}$ using the chain rule of probability:

$$\Pr(\boldsymbol{x}) = \Pr(x_1) \cdot \Pr(x_2|x_1) \cdots \Pr(x_T|x_1,\ldots,x_{T-1}). \quad (1)$$

By learning each of the $T$ conditional distributions with a neural network, AR models such as GPT (Radford et al., 2019; Brown et al., 2020; OpenAI, 2023) and State Space Models (Gu & Dao, 2023; Fu et al., 2022) achieved state-of-the-art performance in modeling high-dimensional sequences like text and protein (Nguyen et al., 2024).

However, AR modeling requires the context to be provided contiguously at the beginning, limiting its applicability for tasks that demand more flexible context handling, such as DNA imputation and protein sequence infilling.

In contrast, non-autoregressive (NAR) modeling seeks to learn conditional distributions given arbitrary contexts. Formally, for a subset of context variables $\mathbf{X}_C$ with $C \subset [T]$, NAR models aim to encode the conditional distribution $\Pr(\mathbf{X}_R|\boldsymbol{x}_C)$ of the remaining variables $\mathbf{X}_R$, where $R := [T]\backslash C$. Here $[T]$ denotes the set of positive integers up to $T$. Compared to AR models, the flexibility of NAR models makes them adaptable to a broader range of conditional generation tasks (Li et al., 2022; Han et al., 2023).

### 2.2. Transformer Models

The Transformer (Vaswani et al., 2017) architecture is commonly used to implement generative models for both AR and NAR generation. Given a sequence of tokens $\{x_t\}_{t=1}^T$, the model initially maps each token $x_t$ into a continuous embedding $\boldsymbol{h}_t^0 \in \mathbb{R}^d$, where $d$ denotes the dimensionality of the embedding space. The embeddings are then iteratively transformed through a stack of $L$ Transformer blocks:

$$\forall l \in \{1,\ldots,L\}, \quad \boldsymbol{h}_{1:T}^l = \texttt{Block}_l(\boldsymbol{h}_{1:T}^{l-1}),$$

where $\boldsymbol{h}_t^l$ represents the embedding at position $t$ in layer $l$ and $\boldsymbol{h}_{1:T}^l := \{\boldsymbol{h}_t^l\}_{t=1}^T$ denotes all embeddings in the layer.

Omitting design details such as the use of layer normalization and residual connections, each Transformer block consists of two modules: a feed-forward neural network (FFN) and an attention module (Attn), which can be expressed as $\boldsymbol{h}_{1:T}^l = \texttt{FFN}_l\big(\texttt{Attn}_l(\boldsymbol{h}_{1:T}^{l-1})\big)$.

(a) The likelihood of "The cat is cute" **differ** when computed in different orders with SEDD.

(b) **LL difference** between the best and the worst order for length-5 sequences.

Compute probability in the forward order

$\log \Pr(X_1 = \texttt{The})$ $\quad = -6.08$
$+$
$\log \Pr(X_2 = \texttt{cat} \,|\, x_1 = \texttt{The})$ $\quad = -11.61$
$+$
$\log \Pr(X_3 = \texttt{is} \,|\, \mathbf{x}_{1:2} = \texttt{The cat})$ $\quad = -3.58$
$+$
$\log \Pr(X_4 = \texttt{cute} \,|\, \mathbf{x}_{1:3} = \texttt{The cat is})$ $= -8.71$
$\parallel$
$\log \Pr(\mathbf{X}_{1:4} = \texttt{The cat is cute})$ $\quad = \boxed{-29.98}$

Compute probability in the reverse order

$\log \Pr(X_4 = \texttt{cute})$ $\quad = -7.22$
$+$
$\log \Pr(X_3 = \texttt{is} \,|\, x_4 = \texttt{cute})$ $\quad = -7.10$
$+$
$\log \Pr(X_2 = \texttt{cat} \,|\, \mathbf{x}_{3:4} = \texttt{is cute})$ $\quad = -7.34$
$+$
$\log \Pr(X_1 = \texttt{The} \,|\, \mathbf{x}_{2:4} = \texttt{cat is cute})$ $= -11.71$
$\parallel$
$\log \Pr(\mathbf{X}_{1:4} = \texttt{The cat is cute})$ $\quad = \boxed{-33.38}$

$\boxed{-29.98} \neq$ ... $= \boxed{-33.38}$

Figure 2: **Different conditional queries do not respect the same joint distribution in NAR models.** (a) The log-likelihoods (LLs) of the sequence "The cat is cute" differ depending on whether SEDD is queried in the forward order or the reverse order. (b) Histogram of the difference between the highest and the lowest LLs of 2,000 length-5 sequences from WikiText103 when queried in all possible orders.

While the FFN is applied independently to embeddings at each position $t$, the attention module captures dependencies between different token variables. Specifically, a learnable function $\texttt{score}_l$ first computes a score value $s_{t,t'}^l := \texttt{score}_l(\boldsymbol{h}_t^{l-1}, \boldsymbol{h}_{t'}^{l-1})$ for each pair of token positions $(t, t')$ using their respective embeddings. The scores are then normalized with the softmax function:

$$\forall t, t' \in \{1, \ldots, T\}, \quad \hat{s}_{t,t'}^l = \frac{m_{t,t'}^l \cdot \exp(s_{t,t'}^l)}{\sum_{t''} m_{t,t''}^l \cdot \exp(s_{t,t''}^l)},$$

where $\mathbf{m}_l := \{m_{t,t'}^l\}_{t,t'} \in \{0,1\}^{T \times T}$ is a mask that specifies the allowed dependencies between token positions.

The normalized scores $\{\hat{s}_{t,t'}^l\}_{t,t'}$ are then used to weigh the embeddings, with the weighted sum of these embeddings forming the final output of the attention module. Specifically, the output at position $t$ is computed as:

$$\texttt{Attn}_l(\boldsymbol{h}_{1:T}^{l-1})_t = \sum_{t'} \hat{s}_{t,t'}^l \cdot g_l(\boldsymbol{h}_{t'}^{l-1}),$$

where $g_l : \mathbb{R}^d \to \mathbb{R}^d$ is a learnable mapping. When implementing AR models with Transformers, causal attention masks are used to ensure each token only receives context from previous tokens:

$$\forall l \in \{1, \ldots, L\}, \quad m_{t,t'}^l = \begin{cases} 1 & \text{if } t' \leq t, \\ 0 & \text{otherwise.} \end{cases} \quad (2)$$

The output $\boldsymbol{h}_t^L$ of the final layer at position $t$ is transformed to predict the conditional distribution $\Pr(X_t | \boldsymbol{x}_{1:t-1})$ (cf. Eq. (1)) through a learnable mapping.

In contrast, NAR modeling is commonly achieved by setting unobserved input tokens to a special placeholder token <MASK>, while configuring the attention mask to allow full visibility of all inputs (Devlin et al., 2018; Lou et al., 2023; Sahoo et al., 2024). Specifically, given contexts $\boldsymbol{x}_C$, all tokens in $\mathbf{X}_R$ with $R := [T] \backslash C$ are assigned the <MASK> token, and the final hidden embedding $\boldsymbol{h}_t^L$ is used to capture $\Pr(X_t | \boldsymbol{x}_C)$ for every $t \in R$.

## 3. Query Generalization in NAR Models

As shown in Figure 1, the strong unconditional generation performance of SoTA NAR models does not ensure high-quality conditional generation, as different conditional queries from the same model do not align with a consistent joint distribution. For example, for any text distribution, the log-likelihood (LL) of the sequence $\boldsymbol{x}_{1:4} :=$ "The cat is cute" should remain the same regardless of how we break down the joint probability into conditional probabilities following the chain rule. However, as shown in Figure 2(a), the LLs computed with SEDD (Lou et al., 2023) in the forward and the reverse order are different, which indicates that the model does not follow the same underlying joint distribution when prompted with different conditional queries. Figure 2(b) further illustrates the prevalence of such inconsistency by measuring the gap between the highest and lowest LLs of 2,000 length-5 sequences from WikiText103 (Merity et al., 2022), evaluated across all possible orders.

As discussed in Section 2.2, tokens not given as evidence to NAR models are often represented by a special <MASK> token. In such cases, generalization to new queries means generalizing to inputs with different <MASK> distribution.

Existing models use different mask strategies during training, which means they are trained to predict certain types of conditional queries, i.e., $\Pr(\cdot | \boldsymbol{x}_C)$ for specific subsets $C \subset [T]$. For instance, discrete diffusion models (Austin et al., 2021) and BERT (Devlin et al., 2018) sample $C$ uniformly at random while T5 (Raffel et al., 2020) sample random spans of variables. However, since it is infeasible to cover all conditional queries during training, NAR models have to achieve robust query generalization to excel at conditional generation.

## 4. Tractable Transformers

The standard Transformer architecture introduced in Section 2.2 is highly expressive and scalable thanks to its attention modules that enable information transformation

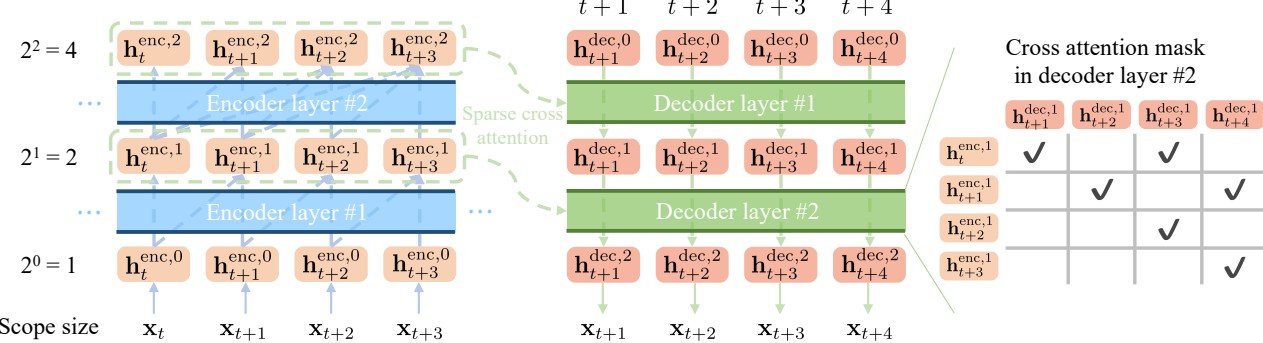

Figure 3: **Tractable Transformers (Tracformers)** use an encoder to learn feature embeddings (i.e., $h_t^{\text{enc},l}$) with context (denoted as scope on the left) sizes ranging from 1 to $2^L$, where $L$ is the number of layers. The encoder features are then fed to a decoder for conditional generation. Each decoder layer contains a feed-forward neural network and a cross-attention layer to collect information from corresponding encoder features. Attention layers in both the encoder and the decoder have special sparse patterns (see Secs. 4.1 and 4.2).

between arbitrary pairs of tokens. However, this global attention mechanism can hinder generalization to unseen conditional queries due to its sensitivity to changes in the distribution of the <MASK> token. In contrast, restricting the context of features $h_t^l$ to local neighborhoods (i.e., makes $h_t^l$ only depends on a small subset of variables) improves robustness across different conditional queries since local features are invariant to changes of mask tokens outside their respective context windows. However, using local features significantly restricts the model's expressiveness.

We propose an encoder-decoder architecture that effectively leverages both local and global features to achieve robustness against query changes while preserving the expressiveness of global context modeling. As shown in Figure 3, the encoder learns feature embeddings with exponentially increasing context lengths in different layers by a sparse attention mechanism. The encoder embeddings are then fed to a cross-attention-only decoder Transformer, which independently predicts the conditional probability of each token. In the following, we introduce the backbone structure of the encoder (Sec. 4.1) and the decoder (Sec. 4.2).

### 4.1. Multi-Scope Encoder

Following Section 2.2, we define $h_t^{\text{enc},l}$ as the feature embeddings at position $t$ of the $l$-th encoder layer (Fig. 3). Define the *variable scope* (or *scope*) $\phi_t^l \subseteq [T]$ of $h_t^{\text{enc},l}$ as the set of variables that contribute to the computation of $h_t^{\text{enc},l}$. For example, with full attention, the scope of every embedding spans the entire input sequence: $\forall t$ and $l, \phi_t^l = [T]$; when using causal attention masks in Equation (2), we have $\phi_t^l = [t]$ for every $l$ and every $t$.

The encoder layers implement a Multi-Scope Self-Attention (MSSA) mechanism such that the size of the variable scopes (i.e., $|\phi_t^l|$) grows exponentially with the layer index $l$. This enables earlier layers to capture fine-grained local semantics,

while later layers encode broader contextual information and more abstract representations. Figure 3 provides an example multi-scope encoder with base 2. Specifically, the scope of each embedding $h_t^{\text{enc},l}$ is the set of variables whose distance from the left of $X_t$ is smaller than $2^l$:

$$\phi_t^l = \{t' : t' \geq 1, 0 \leq t - t' < 2^l\}. \tag{3}$$

MSSAs can be implemented by applying sparse attention masks to a standard attention module as introduced in Section 2.2. Specifically, the base-2 scope pattern in Figure 3 can be achieved using the following attention masks for each $l \in [L]$, and $t, t' \in [T]$:

$$m_{t,t'}^{\text{enc},l} = \begin{cases} 1 & \text{if } t - 2^{l-1} \leq t' \leq t, \\ 0 & \text{otherwise.} \end{cases} \tag{4}$$

Moreover, we note that only $\mathcal{O}(T)$ 1s in every attention mask $\mathbf{m}_l^{\text{enc}} \in \{0, 1\}^{T \times T}$ are needed to get the desired exponentially increasing scope pattern. For example, when the base is 2, instead of attending to every token in the range $t - 2^{l-1} \leq t' \leq t$ (Eq. (4)), we only need to attend to the two tokens at the boundary, i.e., $m_{t,t'}^{\text{enc},l} = 1$ when $t' = t$ or $t' = t - 2^{l-1}$, to obtain the desired variable scope in Equation (3).[1] This enables linear time (w.r.t. sequence length) implementations of the encoder, effectively avoiding the quadratic computation overhead of standard Transformers.

To control this computational complexity, we define a hyperparameter $N_{\max}$ ($N_{\max} \geq 2$) that specifies the maximum number of embeddings a given token can attend to using the attention mask $\mathbf{m}_l^{\text{enc}}$. Specifically, if the range $t - 2^{l-1} \leq t' \leq t$ in Equation (4) contains no more than $N_{\max}$ embeddings, all embeddings in the range are attended to

---

[1]According to the definition in Equation (3), we have that $\phi_{t-2^{l-1}}^{l-1} = \{t' : t' \geq 1, 0 \leq t - 2^{l-1} - t' < 2^{l-1}\}$ and $\phi_t^{l-1} = \{t' : t' \geq 1, 0 \leq t - t' < 2^{l-1}\}$. Taking the union of the two sets leads to $\phi_t^l := \phi_{t-2^{l-1}}^{l-1} \cup \phi_t^{l-1} = \{t' : t' \geq 1, 0 \leq t - t' < 2^l\}$.

(i.e., the corresponding mask values are 1). Otherwise, the attention mask selects exactly $N_{\max}$ embeddings from this range by sample linearly and round to the nearest integer.

To ensure the existence of encoder embeddings with full context (i.e., its variable scope is $[T]$), we choose the number of layers $L$ to be greater than or equal to $\log_2 T$.

## 4.2. Decoder

Recall from the beginning of Section 4 that the decoder combines local and global features computed in the encoder to predict distributions. As illustrated in Figure 3, the decoder consists of $L$ Transformer blocks, each containing a feed-forward neural network (FFN) and a cross-attention layer. Formally, given inputs $\boldsymbol{h}_{1:T}^{\text{dec},l-1}$ to the $l$-th decoder block, the outputs $\boldsymbol{h}_{1:T}^{\text{dec},l}$ are computed as

$$\boldsymbol{h}_{1:T}^{\text{dec},l} = \text{FFN}_l\big(\text{CrossAttn}_l(\boldsymbol{h}_{1:T}^{\text{dec},l-1}, \boldsymbol{h}_{1:T}^{\text{enc},L-l+1})\big),$$

where in the cross-attention operation, each decoder embedding $\boldsymbol{h}_t^{\text{dec},l-1}$ ($\forall t \in [T]$) attends to the outputs $\boldsymbol{h}_{1:T}^{\text{enc},L-l+1}$ of the $(L-l+1)$-th encoder layer. Intuitively, the decoder first cross-attend to encoder embeddings that encode global information (i.e., those with large scope sizes) before focusing on embeddings encoding local information (i.e., those with small scope sizes). This progressive refinement mirrors human language generation, where high-level intent is first structured before being articulated into detailed expressions.[2]

Attention masks are used in cross-attention layers to ensure each decoder embedding only depends on desired input variables. For example, when given the context $\boldsymbol{x}_C$, since the $t$-th decoder embedding $\boldsymbol{h}_t^{\text{dec},l}$ is used to predict $\Pr(X_t|\boldsymbol{x}_C)$, $\boldsymbol{h}_t^{\text{dec},l}$ should not attend to encoder embeddings whose variable scope contain variables not in $X_C$.

Formally, define $m_{t,t'}^{\text{dec},l}$ as the mask deciding whether the $t$-th embedding of the input to the $l$-th decoder block (i.e., $\boldsymbol{h}_t^{\text{dec},l-1}$) can attend to the $t'$-th embedding in the corresponding encoder layer (i.e., $\boldsymbol{h}_{t'}^{\text{enc},L-l+1}$).

Different generation paradigms can be achieved using specific decoder masks $\mathbf{m}_{\text{dec}}^l := \{m_{t,t'}^{\text{dec},l}\}_{t,t'}$ For example, to achieve AR modeling with Tracformers, we set $m_{t,t'}^{\text{dec},l}=1$ if $t' < t$ and zero out all remaining mask entries. Additionally, we observe that attending to fewer encoder tokens in the initial decoder layers has little impact on the performance. Therefore, to improve computational efficiency, we scale the sparsity of the cross-attention in proportion to the scope length of the corresponding encoder layer. Formally, the

---

[2]For example, Wan et al. (2025) highlights the effectiveness of a top-down hierarchical scheme in long text generation.

decoder attention mask is defined as:

$$m_{t,t'}^{\text{dec},l} = \begin{cases} 1 & \text{if } t' < t \text{ and } t' \equiv t-1 \,(\text{mod } 2^{L-l+1}), \\ 0 & \text{otherwise.} \end{cases} \quad (5)$$

### 4.3. Tractable Transformers for NAR Generation

In this section, we demonstrate how to apply Tracformers to two common NAR generation paradigms—contextual autoregressive generation and arbitrary-context generation.

**Contextual AR Generation.** Contextual AR (CAR) generation refers to a scenario in which, when predicting a variable $X_t$, the model has access to all preceding tokens $\boldsymbol{x}_{1:t-1}$ and a subset of future tokens. This paradigm is used when an arbitrary context $\boldsymbol{x}_C$ is provided, and the model is tasked with autoregressively sampling all remaining tokens.

To implement Tracformer for CAR generation, we use two encoders, a *prefix encoder* and a *suffix encoder*, which capture information from preceding and succeeding tokens, respectively. As shown in Figure 4(a), the prefix encoder processes the original sequence during training since preceding tokens are always available in CAR generation. In contrast, the suffix encoder observes only the tokens in a chosen context set $\boldsymbol{x}_C$, while the remaining tokens are represented by the <MASK> token.

The prefix encoder uses the sparse attention mask described in Section 4.1 (i.e., the sparse version of Eq. (4)), where the scope of each feature $\boldsymbol{h}_t^{\text{enc},l}$ includes the $2^l$ preceding variables, including $X_t$. We use similar attention masks in the suffix encoder such that the scope of each feature $\boldsymbol{h}_t^{\text{enc},l}$ covers the $2^l$ succeeding variables starting from $X_t$. See Appendix A for a formal description. We use the masks defined in Equation (5) for cross-attention between the decoder and the prefix encoder, as the attended features contain only prefix token information. Analogously, the suffix encoder uses the following cross-attention masks to ensure the decoder receives only suffix information from it:

$$m_{t,t'}^{\text{dec},l} = \begin{cases} 1 & \text{if } t' > t \text{ and } t' \equiv t+1 \,(\text{mod } 2^{L-l+1}), \\ 0 & \text{otherwise.} \end{cases} \quad (6)$$

As illustrated in Figure 4(a), given context $\boldsymbol{x}_C$, the $t$-th output feature embedding of the decoder is used to predict the distribution $\Pr(X_t|\boldsymbol{x}_{C_t})$, where $C_t := C \cup [t-1]$. The overall training loss for the CAR generation model is

$$L(\boldsymbol{\theta}) = -\mathbb{E}_{\boldsymbol{x} \sim \mathcal{D}, C \sim \mathcal{P}_C}\left[\sum_{t \notin C} \log \Pr_{\boldsymbol{\theta}}(x_t|\boldsymbol{x}_{C_t})\right], \quad (7)$$

where $\boldsymbol{\theta}$ is the set of learnable parameters, $\mathcal{D}$ is a dataset, and $\mathcal{P}_C$ is a mask strategy used to sample the context set $C$.

**Arbitrary-Context Generation.** In arbitrary-context (AC) generation, the model is tasked to predict the distribution

(a) Tracformer for **contextual AR (CAR)** generation.

(b) Tracformer for **arbitrary-context (AC)** generation.

Figure 4: **Tracformers for contextual AR (CAR) and arbitrary-context (AC) generation.** In both cases, we use a prefix encoder and a suffix encoder to capture prefix and suffix information for every token, respectively. Model parameters of the two encoders are shared. The decoder receives information from features generated by both encoders. Specifically, the decoder attention masks ensure the decoder only acquires information of preceding (resp. succeeding) variables from the prefix (resp. suffix) encoder. During training, <MASK> tokens are given to both encoders if the model is trained for AC generation. When training Tracformer for CAR generation, the prefix encoder always receives all inputs as preceding tokens $\boldsymbol{x}_{1:t-1}$ are assumed to be given when sampling $x_t$.

of $X_t$ given arbitrary context, which can be used to generate missing tokens in an arbitrary order. Analogous to Equation (7), the objective of AC generation task is

$$L(\boldsymbol{\theta}) = -\mathbb{E}_{\boldsymbol{x}\sim\mathcal{D},C\sim\mathcal{P}_C}\left[\sum_{t\notin C}\log\Pr_{\boldsymbol{\theta}}(x_t|\boldsymbol{x}_C)\right], \quad (8)$$

As shown in Figure 4(b), AC generation paradigm can be implemented by the same model described for CAR generation, with the only difference that inputs to both the prefix encoder and the suffix encoder use the mask token if a token does not exist in the context. Correspondingly, the $t$-th output feature embedding predicts the distribution $\Pr(X_t|\boldsymbol{x}_c)$.

**Training and Inference Efficiency.** Thanks to its sparse attention modules in both the encoder and the decoder, Tracformers enjoy efficient training and inference. In particular, KV-caching (Pope et al., 2023) can be used to amortize inference cost. See Appendix E for a detailed discussion.

## 5. Related Work

**NAR Modeling Techniques.** BERT (Devlin et al., 2018; Warner et al., 2024) is one of the first Transformer models designed for NAR generation. It proposes to use a special mask token to indicate unknown tokens and task the model to predict them given the observed tokens. Built on top of this mask prediction principle, discrete diffusion models (Austin et al., 2021) improve NAR generation performance by designing better learning objectives (Campbell et al., 2022; Lou et al., 2023; Sahoo et al., 2024) and mask strategies (Shi et al., 2024). Instead of recovering sequences from mask tokens, some discrete diffusion models learn to recover from uniformly sampled sequences (Lou et al., 2023). Another thread of work incorporates autoregressive or semi-autoregressive biases to the denoising process of diffusion models, intending to combine the expressiveness of autoregressive modeling and the ability to perform NAR generation (Chen et al., 2024; Han et al., 2023).

**Architectures for NAR Modeling.** Decoder-only transformers with full attention are the most widely adopted architecture for NAR modeling. Many SoTA discrete diffusion models use these models. Additionally, bidirectional autoregressive modeling, exemplified by models like BART (Lewis et al., 2020) and MASS (Song et al., 2019), represents an intermediate approach that incorporates bidirectional context while preserving the left-to-right autoregressive generation process. Sun et al. (2023) developed a Transformer-based architecture for a subclass of discrete diffusion models. Liu et al. (2025) and Xu et al. (2025) combine diffusion models with other deep generative models, such as AR models and energy-based models.

## 6. Experiment

In this section, we aim to empirically evaluate Tracformer's effectiveness in both conditional and unconditional generation. Specifically, our experiments are designed to answer two key questions: (i) How does Tracformer compare to other NAR architectures in terms of conditional generation performance? (ii) Can Tracformer scale effectively and outperform existing SoTA generative models in both conditional and unconditional tasks? To this end, we conduct two sets of experiments: In Section 6.1, we compare Tracformer with a range of NAR architectures on WikiText (Merity et al., 2022), LAMBADA (Paperno et al., 2016), and One Billion Words (1BW) (Chelba et al., 2013) datasets to evaluate its performance across diverse conditional queries. In Section 6.2, we scale Tracformer to OpenWebText (Gokaslan & Cohen, 2019) and benchmark it against SoTA discrete diffusion models, focusing on zero-shot conditional and unconditional performance. These experiments comprehensively evaluate Tracformer's advantages and its potential to serve as a more effective backbone for NAR generation. Note that the perplexity numbers of diffusion model baselines are computed using the corresponding ELBO since the exact log-likelihood is intractable to compute.

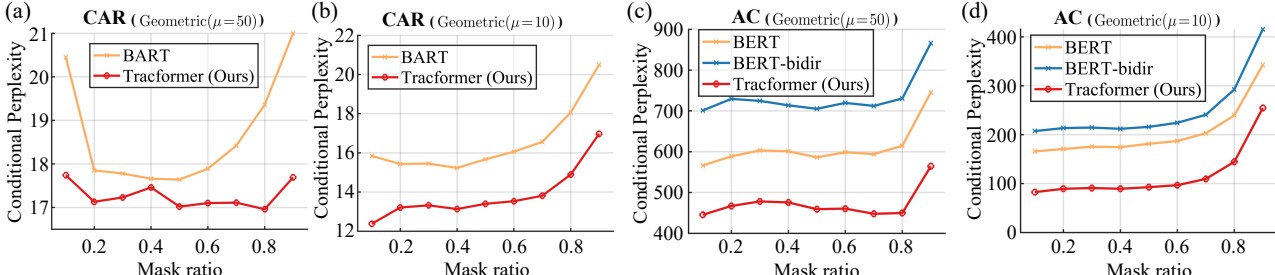

Figure 5: Validation conditional perplexity (↓) of CAR/AC generation tasks with varying masking strategies, evaluated on the WikiText103 validation set with a sequence length of 1024. (a) and (b) show CAR generation with span lengths sampled from Geometric($\mu = 50$) and Geometric($\mu = 10$), respectively. (c) and (d) correspond to AC generation with the same span length distributions as in CAR. Across all mask ratios and span lengths, Tracformer consistently outperforms all baselines, demonstrating strong generalization capabilities.

Table 1: Evaluation of CAR text infilling performance using the MAUVE and BERT Score with different mask ranges. Higher scores indicate better performance. Tracformer consistently outperforms BART under all conditions.

| Mask ranges | MAUVE (↑) | | BERT Score (↑) | |
|---|---|---|---|---|
| | Tracformer | BART | Tracformer | BART |
| [0.25,0.75] | **0.960** | 0.951 | **0.464** | 0.414 |
| [0.5,1.0] | **0.114** | 0.016 | **0.488** | 0.386 |
| [0.1,0.4] & [0.6,0.9] | **0.931** | 0.889 | **0.370** | 0.306 |
| [0,0.4] & [0.5,0.8] | **0.946** | 0.940 | **0.274** | 0.216 |
| [0,0.25] & [0.75,1.0] | **0.177** | 0.063 | **0.457** | 0.401 |
| [0,0.1] & [0.2,0.5] & [0.7,1.0] | **0.107** | 0.023 | **0.286** | 0.208 |

### 6.1. Comparison of Architectures for NAR Modeling

Recall from Section 4 that we propose Tracformer's encoder-decoder architecture as an effective approach to leveraging both local and global features, enabling robust performance across varying queries. To empirically validate this, we compare Tracformer with baseline models employing different transformer architectures on identical NAR modeling tasks, focusing on their conditional generalization capabilities.

**Baselines.** As discussed in Section 4.3, Tracformer supports two distinct NAR generation modes: contextual AR (CAR) generation and arbitrary-context (AC) generation, each requiring different baseline comparisons. For CAR generation, we use BART (Lewis et al., 2020) as a baseline, as it represents a widely adopted encoder-decoder architecture for CAR tasks. BART's encoder captures bidirectional global context, while its autoregressive decoder generates outputs sequentially via cross-attention. This fundamental design principle is shared by many CAR models, such as MASS (Song et al., 2019) and PALM (Bi et al., 2020), making BART a strong representative baseline.

For AC generation, we compare Tracformer against two baselines: (i) BERT (Devlin et al., 2018), which is the predominantly used backbone for modern diffusion models, and (ii) BERT-bidir, a variant of BERT that incorporates both forward and reverse AR encoders using dense self-attention. These baselines provide a contrast between standard bidirectional encoding and autoregressive modeling in

AC generation tasks (see Appx. B.2 for details).

**Training Setup.** For the CAR task, Tracformer and BART are both trained using the CAR objective defined in Equation (7). During training, we set the mask strategy (i.e., $\mathcal{P}_C$) to sample spans whose lengths follow a geometric distribution with mean $\mu = 50$ and apply a total mask ratio of 50% (see Appx. B.1). All models are trained on the WikiText103, LAMBADA, and 1BW. We present the results of WikiText103 in the paper and defer other results to Appendix B.3. The experiment setup for the AC generation task follows the CAR task, where the training objective is defined in Equation (8). See Appendix B.2 for more details.

**Empirical Insights from the CAR Generation Results.** We evaluate both models' generalization capabilities on CAR tasks through two specific tests: ratio generalization and span generalization. For ratio generalization evaluation, the span masking strategy remains consistent with the training setup, where the span length is sampled from Geometric($\mu = 50$), but the total mask ratio is varied between 0.1 and 0.9. Figure 5(a) shows the conditional perplexity (PPL) for both models on the WikiText103 validation set. The results exhibit a U-shape, where performance improves as the mask ratio approaches the training mask ratio. Tracformer consistently outperforms BART across all mask ratios. The performance gap between the two models is small near the training mask ratio, but BART's PPL increases as the mask ratio deviates further, whereas Tracformer maintains robust generalization performance.

To evaluate span generalization performance, we alter the mean span lengths of the adopted span masking strategy during inference, resulting in a mismatch $\mathcal{P}_C$ between training and evaluation. Figure 5(b) illustrates the conditional PPL when the mean span length is reduced to 10. As expected, shorter span lengths reduce the overall task difficulty, resulting in lower PPL for all models. However, Tracformer demonstrates significantly better performance improvements compared to BART, highlighting its ability to generalize effectively to unseen masking patterns. We also include Geometric($\mu = 3$) span mask results in Figure 7(a)

Table 2: Zero-shot conditional perplexity on WikiText103 and 1BW using six fixed prompts. Tracformer operates in the CAR generation mode, while the conditional PPLs of SEDD and MDLM are computed following the derivation in Appendix G. Tracformer consistently outperforms state-of-the-art discrete diffusion models SEDD and MDLM across various masking ranges.

| Mask ranges | WikiText103 | | | 1BW | | |
|---|---|---|---|---|---|---|
| | Tracformer | SEDD | MDLM | Tracformer | SEDD | MDLM |
| [0.25,0.75] | **29.38** | 37.83 | 30.36 | **41.98** | 53.72 | 55.19 |
| [0.1,0.4] & [0.6,0.9] | **29.67** | 37.17 | 30.55 | **38.52** | 49.38 | 49.48 |
| [0,0.4] & [0.5,0.8] | **34.73** | 43.86 | 35.84 | **45.56** | 58.22 | 58.41 |
| [0,0.25] & [0.75,1] | **37.73** | 46.65 | 40.11 | **49.14** | 66.96 | 66.16 |
| [0.2,0.3] & [0.4,0.6] | 24.31 | 28.31 | **22.97** | **31.76** | 38.76 | 38.97 |
| [0.1,0.9] | **33.88** | 44.92 | 36.69 | **43.98** | 57.96 | 59.14 |

of Appendix B.3, where a similar trend is observed.

Beyond perplexity, we assess the quality of generated text in CAR tasks using six fixed mask ranges (i.e., text infilling). Metrics such as MAUVE (Pillutla et al., 2021) and BERT Score (Zhang et al., 2020) are used to quantify the similarity between the generated and original text. See Appendix B.3 for further evaluation details. Table 1 presents the results, where Tracformer consistently achieves higher scores than all baselines, demonstrating Tracformer's ability to produce high-quality text under diverse conditions.

**Empirical Insights from the AC Generation Results.**
As shown in Figure 5(c) and (d), Tracformer achieves significantly lower conditional perplexity than both baselines across all mask ratios and span lengths, demonstrating superior generalization performance consistently. Additionally, we conduct an ablation study on Tracformer's multi-scope attention, with a detailed analysis provided in Appendix D.

### 6.2. Benchmarking Against SoTA NAR Models

In this section, we test the scalability of Tracformer by training it on the OpenWebText (Gokaslan & Cohen, 2019) dataset and comparing its performance against various SoTA generative models of similar scale, including GPT-2 (Radford et al., 2019) and SoTA discrete diffusion models (Lou et al., 2023; Sahoo et al., 2024; Austin et al., 2021; Gulrajani & Hashimoto, 2024). Tracformer is trained using the CAR generation objective defined in Equation (7), with similar masking strategies $\mathcal{P}_C$ to Section 6.1 (see Appx. C). For evaluation, we benchmark zero-shot conditional and unconditional performance on multiple datasets.

For conditional generation, we compare Tracformer with SoTA discrete diffusion models, SEDD and MDLM. Table 2 presents the zero-shot conditional PPL results using fixed masking strategies, where Tracformer consistently outperforms both baselines across all mask ranges. To further assess generalization, we evaluate varied masking strategies, with Figure 6 reporting results from ratio-generalization and span-generalization tests. Specifically, we consider

Table 3: Model size, number of non-embedding parameters, and number of trained tokens for the models evaluated in this paper. Tracformer has a smaller full model size and fewer non-embedding parameters compared to SEDD and MDLM.

| Model | Full Model Size | #Non-Embed Params | #Trained Tokens |
|---|---|---|---|
| GPT2 | 124M | 85M | - |
| SEDD | 169M | 90M | 210B |
| MDLM | 169M | 90M | 524B |
| Tracformer | 109M | 79M | 295B |

Table 4: Zero-shot unconditional perplexity across various datasets. Tracformer, despite its smaller size, achieves competitive or superior performance compared to larger models.

| Model | WikiText103 | WikiText2 | Lambada | PTB | 1BW |
|---|---|---|---|---|---|
| GPT-2 (124M) | 41.60 | 42.32 | **45.04** | 138.43 | 75.20 |
| D3PM (169M) | 75.16 | 77.28 | 93.47 | 200.82 | 138.92 |
| PLAID (169M) | 50.86 | 51.80 | 57.28 | 142.60 | 91.12 |
| SEDD (169M) | 40.62 | 41.84 | 50.92 | 114.24 | 79.29 |
| MDLM (169M) | **37.01** | **36.75** | 48.46 | **96.40** | 67.94 |
| Tracformer (109M) | 43.27 | 43.82 | 58.10 | 166.10 | **51.34** |

two span length distributions: Geometric($\mu = 10$) and DLogistic($\mu = 15, \sigma = 3$). On 1BW, Tracformer consistently outperforms both SEDD and MDLM across all mask ratios. On WikiText103, MDLM shows a slight advantage at low mask ratios, but as the masking ratio increases, Tracformer achieves the best performance, demonstrating stronger generalization under more challenging conditions.

Notably, despite its smaller model size (Tab. 3), Tracformer not only excels in fixed-mask conditional PPL (Tab. 2) but also adapts better to diverse masking distributions.

For unconditional generation, Table 4 reports the zero-shot unconditional PPL across various datasets. Tracformer remains highly competitive, achieving results comparable to or better than larger models. We additionally include text samples generated by Tracformer in Appendix F for both conditional and unconditional generation. These findings further reinforce our earlier conclusion: while SoTA NAR diffusion models exhibit strong unconditional PPL, their inability to generalize effectively to conditional queries unseen during training limits their broader applications. Tracformer, with its robust performance in both conditional and unconditional tasks, demonstrates its potential as a scalable and versatile generative model.

## 7. Conclusion and Limitations

We propose Tracformer, a Transformer-based architecture for flexible and generalizable conditional generation. Through extensive experiments, we demonstrate that Tracformer's multi-scope attention mechanism and specialized encoder-decoder design enable robust conditional generation performance. However, due to resource limitations, we only train Tracformer at the GPT-2 (base) scale. While

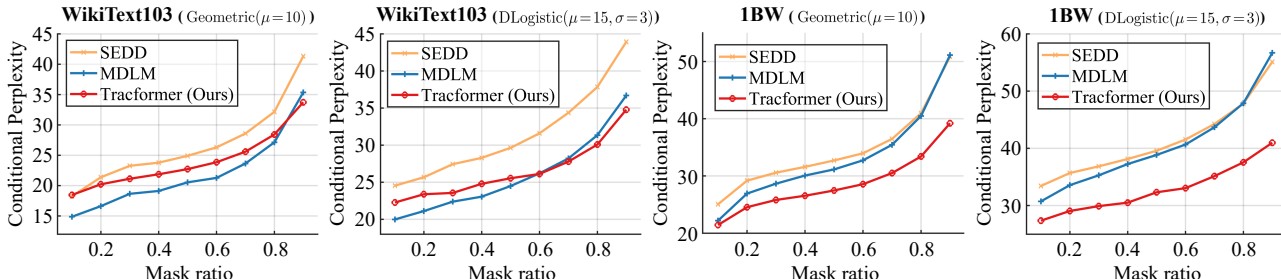

Figure 6: Conditional perplexity under varied masking strategies, evaluating ratio-generalization and span-generalization. We evaluate two different span length distributions: Geometric($\mu = 10$) and DLogistic($\mu = 15, \sigma = 3$). Tracformer demonstrates superior conditional generalization capabilities compared to SEDD and MDLM across different masking strategies.

our current results already establish Tracformer as a highly promising NAR architecture, future work will focus on scaling the model further to fully explore its potential. Moreover, given that AC generation closely aligns with one-step diffusion training, Tracformer could serve as a strong backbone for modern diffusion models. Future research will investigate this direction, leveraging Tracformer's capabilities to enhance diffusion-based generative frameworks.

## Impact Statement

This paper presents work whose goal is to advance the field of Machine Learning. There are many potential societal consequences of our work, none which we feel must be specifically highlighted here.

## Acknowledgements

This work was funded in part by the National Science and Technology Major Project (2022ZD0114902), the DARPA ANSR program under award FA8750-23-2-0004, the DARPA CODORD program under award HR00112590089, the DARPA SAFRON program under award HR00112530141, the Deutsche Forschungsgemeinschaft (DFG, German Research Foundation) under Germany's Excellence Strategy - EXC 2075 – 390740016, NSF grant #IIS-1943641, and gifts from Adobe Research, Cisco Research, and Amazon. We acknowledge the support of the Stuttgart Center for Simulation Science (SimTech). MN thanks IMPRS-IS (International Max Planck Research School for Intelligent Systems) for the support. Approved for public release; distribution is unlimited.

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

# A. Additional Details of Tracformers

This section introduces the design details of Tracformers.

**Sparse Attention Masks of the Encoder.**   As described in Section 4.3, a prefix encoder and a suffix encoder are used for both CAR and AC generation. Specifically, the attention mask for each layer is the sparsified version of Equation (4) such that each token attends to at most $N_{\max}$ tokens. Formally, the attention map $\mathbf{m}_{\text{enc}}^l$ of layer $l \in [L]$ in the prefix encoder is given by (denote $a = t - 2^{l-1}, d = \left\lfloor \frac{2^{l-1}}{N_{\max}-1} \right\rfloor, b = t$):

$$m_{t,t'}^{\text{enc},l} = \begin{cases} 1 & \text{if } t' \in S_t^l, \\ 0 & \text{otherwise,} \end{cases} \quad \text{where } S_t^l = \begin{cases} \{t' \in \mathbb{Z} \mid t - 2^{l-1} \leq t' \leq t, t' \geq 1\}, & \text{if } 2^{l-1} \leq N_{\max}, \\ \{a, a+d, a+2d, \ldots, b\} \cap [T], & \text{if } 2^{l-1} > N_{\max}. \end{cases}$$

As shown in Section 4.3, each feature $\boldsymbol{h}_t^{\text{enc},l}$ in the suffix encoder covers the $2^l$ succeeding variables starting from $X_t$:

$$\phi_t^l = \{t' : t' \leq T, 0 \leq t' - t < 2^l\},$$

which is analogous to Equation (3). Similar to the prefix encoder, we use the following sparse attention mask to implement the suffix encoder (denote $a = t, d = \left\lfloor \frac{2^{l-1}}{N_{\max}-1} \right\rfloor, b = t + 2^{l-1}$):

$$m_{t,t'}^{\text{enc},l} = \begin{cases} 1 & \text{if } t' \in S_t^l, \\ 0 & \text{otherwise,} \end{cases} \quad \text{where } S_t^l = \begin{cases} \{t' \in \mathbb{Z} \mid t \leq t' \leq t + 2^{l-1}, t' \leq T\}, & \text{if } 2^{l-1} \leq N_{\max}, \\ \{a, a+d, a+2d, \ldots, b\} \cap [T], & \text{if } 2^{l-1} > N_{\max}. \end{cases}$$

**Sparse Attention Masks of the Decoder.**   As discussed in Section 4.3, the decoder cross-attends to features in both the prefix encoder and the suffix encoder, using the mask in Equations (5) and (6), respectively.

**Encoder Blocks.**   Each encoder block contains an attention layer and an FFN following the standard design. Skip connections and layer normalization layers are used.

$$\hat{\boldsymbol{h}}_{1:T}^{\text{enc},l} = \boldsymbol{h}_{1:T}^{\text{enc},l-1} + \texttt{Attn}_l(\texttt{LayerNorm}_{l,1}(\boldsymbol{h}_{1:T}^{\text{enc},l-1})),$$
$$\boldsymbol{h}_{1:T}^{\text{enc},l} = \hat{\boldsymbol{h}}_{1:T}^{\text{enc},l} + \texttt{FFN}_l(\texttt{LayerNorm}_{l,2}(\hat{\boldsymbol{h}}_{1:T}^{\text{enc},l})).$$

In the attention layers, we adopt the Rotary positional encoding (Su et al., 2024) to encode information about relative positions between feature embeddings. This is widely adopted in NAR models such as discrete diffusion models (Lou et al., 2023; Sahoo et al., 2024).

For the FFN, we follow the design of GPT-2 and use a two-layer fully connected neural network with GeLU (Hendrycks & Gimpel, 2016) activation. The input and output dimensions are both the embedding dimension $d$ and the latent dimension size is $4 \times d$.

**Decoder Blocks.**   Each decoder block consists of a cross-attention layer and an FFN:

$$\hat{\boldsymbol{h}}_{1:T}^{\text{dec},l} = \boldsymbol{h}_{1:T}^{\text{dec},l-1} + \texttt{CrossAttn}_l(\texttt{LayerNorm}_{l,1}(\boldsymbol{h}_{1:T}^{\text{dec},l-1}), \texttt{LayerNorm}_{l,2}(\boldsymbol{h}_{1:T}^{\text{enc},L-l+1})),$$
$$\boldsymbol{h}_{1:T}^{\text{dec},l} = \hat{\boldsymbol{h}}_{1:T}^{\text{dec},l} + \texttt{FFN}_l(\texttt{LayerNorm}_{l,3}(\hat{\boldsymbol{h}}_{1:T}^{\text{dec},l})).$$

Similar to the encoder, we also adopt the Rotary positional encoding in cross-attention layers. We set the input features to the decoder to zero (i.e., $\boldsymbol{h}_{1:T}^{\text{dec},0} = \boldsymbol{0}$). The FFNs are the same as in the encoder blocks.

# B. Comparison of NAR Architectures in Small-Scale Experiments

### B.1. Training Mask Strategy

The span masking strategy $\mathcal{P}_C$ used in our experiments is implemented using the following algorithm:

---

**Algorithm 1** Span Masking Strategy

---

1: **Input:** Sequence length $L$, mask probability $p$, mean span length $m$, distribution type (Geometric or DLogistic), $\sigma$
2: Initialize $num\_to\_mask = \max(1, \text{round}(p \cdot L))$, $num\_masked = 0$, $blank\_ids = []$, $spans = []$
3: **while** $num\_masked < num\_to\_mask$ **do**
4:     Sample a random $start$ index in $[0, L)$
5:     **if** $start$ overlaps with an existing span in $spans$ **then**
6:         **continue**
7:     **end if**
8:     **if** distribution == Geometric **then**
9:         Sample $span\_length \sim \text{Geometric}(\mu = m)$                      ▷ Geometric distribution with mean $\mu$
10:     **else if** distribution == DLogistic **then**
11:         Sample $span\_length \sim \text{DLogistic}(\mu = m, \sigma = \sigma)$      ▷ Dlogistic distribution with mean $\mu$ and std $\sigma$
12:     **end if**
13:     $end = \min(start + span\_length, L)$
14:     **if** Overlap with existing spans detected **then**
15:         **continue**
16:     **end if**
17:     Append $(start, end)$ to $spans$, update $blank\_ids$ and $num\_masked$
18: **end while**
19: **if** $num\_masked > num\_to\_mask$ **then**
20:     Trim the last span to ensure exact masking ratio
21: **end if**
22: **Return:** $spans, blank\_ids$

---

Following Equation (7), the CAR training process involves applying the span masking strategy to corrupt input sequences and training the model to predict the original tokens based on the provided context. Specifically, at each training iteration, the generate_span_mask function is used to create masked spans within the input sequence. The masking process ensures that 50% of the tokens (based on a mask probability of 0.5) are selected in spans, where the span length is sampled from either a `Geometric` or `DLogistic` distribution. For training in Section 6.1, we use the `Geometric` distribution with a mean span length of $\mu = 50$. The corrupted input sequence, along with a mask indicating the positions of the masked tokens, is then passed to the model. The model is trained to minimize the negative log-likelihood of the original tokens at the masked positions, conditioning on the unmasked tokens as context.

### B.2. Model and Training Configurations

**Training Hyperparameters.** For both CAR and AC training tasks, the sequence length is set to 1024 tokens, with a batch size of 256. The models are optimized using AdamW with $\beta_1 = 0.9$, $\beta_2 = 0.95$, and a weight decay of 0.1. The initial learning rate is set to $6 \times 10^{-4}$ and follows a cosine decay schedule, with 1,000 warmup steps to stabilize the early training phase. The final learning rate is $6 \times 10^{-5}$. Training is conducted for 30,000 steps.

**Hyperparameters of Tracformer.** As listed in Table 6, for both CAR and AC training tasks, Tracformer is implemented with a 10-layer encoder-decoder architecture, maintaining a block size (i.e., maximum sequence length) of 1024 tokens. It utilizes sparse multi-scope attention with a constraint of 16 attended tokens per step, allowing for efficient context aggregation while keeping computational cost manageable. The decoder operates with a maximum stride of 1024 tokens, ensuring global context encoding. The model is configured with 9 attention heads and an embedding dimension of 576. A dropout rate of 0.1 is applied to mitigate overfitting during training.

**Hyperparameters of the Baseline Models.** As the CAR baseline, BART adopts an 8-layer encoder-decoder architecture with 9 attention heads and an embedding dimension of 576. This leads to a total of 138M parameters.

For the AC generation experiments, we use BERT as the encoder-only baseline with full attention, employing 10 layers, 12 attention heads, and an embedding dimension of 768. This configuration ensures a fair comparison by maintaining a similar model size to Tracformer. Additionally, we include BERT-bidir, which encodes bidirectional context using two separate encoders: a classical forward AR encoder with causal attention and a reverse AR encoder. While structurally similar to Tracformer's prefix and suffix encoders, BERT-bidder relies on dense causal attention and lacks the sparse multi-scope self-

Table 5: Training hyperparameters for CAR and AC tasks.

| Parameter | Value |
|---|---|
| *Training Setup* | |
| Sequence length | 1024 tokens |
| Batch size | 256 |
| Training steps | 30,000 |
| *Optimizer* | |
| Optimizer | AdamW |
| $\beta_1$ | 0.9 |
| $\beta_2$ | 0.95 |
| Weight decay | 0.1 |
| *Learning Rate Schedule* | |
| Initial learning rate | $6 \times 10^{-4}$ |
| Final learning rate | $6 \times 10^{-5}$ |
| Learning rate decay | Cosine schedule |
| Warmup steps | 1,000 |

Table 6: Hyperparameters of Tracformer for the Small-scale Experiments.

| Parameter | Value |
|---|---|
| *Structure Parameters* | |
| Block size | 1024 |
| Number of encoder layers | 10 |
| Number of decoder layers | 10 |
| Max attended tokens ($N_{\max}$) | 16 |
| *Attention Parameters* | |
| Number of attention heads | 9 |
| Embedding dimension ($d$) | 576 |
| *Other Parameters* | |
| Dropout rate | 0.1 |

Table 7: Comparison of the model size for small-scale experiments.

| Model | Full Model Size |
|---|---|
| Tracformer | 109M |
| BART | 138M |
| BERT | 149M |
| BERT-bidir | 170M |

attention mechanism that enhances Tracformer's efficiency in handling conditional generation. BERT-bidder is configured with 12 layers, 12 attention heads, an embedding dimension of 768, and a dropout rate of 0.1. After encoding, the final-layer features from both encoders are concatenated and passed through a two-layer MLP to produce the final representation, allowing the model to integrate information from both directions before prediction. The MLP consists of two fully connected layers (config.n_embd is the embedding size of the model):

$$
\begin{aligned}
\text{mlp} = \text{nn.ModuleList}([&\text{nn.Linear}(\text{config.n\_embd} \times 2, \text{config.n\_embd} \times 4), \\
&\text{GELU}(), \\
&\text{nn.Linear}(\text{config.n\_embd} \times 4, \text{config.n\_embd}), \\
&\text{GELU}()])
\end{aligned}
$$

Table 7 shows the parameter count of the adopted baseline models.

### B.3. Additional Evaluation Details and Results

**Conditional PPL under Short Span Masking.**  Figure 7 contains complementary results of Figure 5.

**Metrics for CAR Text Infilling.**  For the CAR text infilling task, a sequence length of 128 is used. To compute MAUVE and BERT scores, we evaluate 1,000 text sequences from the WikiText-103 validation set across all methods. Each prompt generates five samples, resulting in 5,000 generated sequences.

**CAR Text Infilling Results on Other Datasets.**  Text-infilling performance on the 1BW and LAMBADA datasets are given in Table 8.

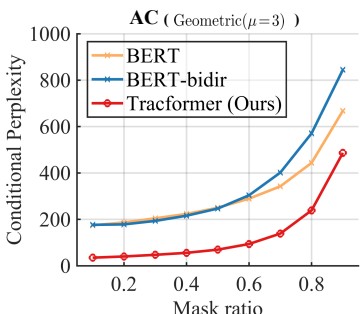
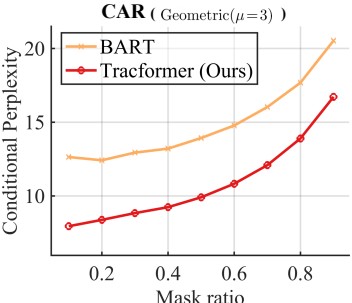

Figure 7: Conditional perplexity (PPL) evaluated on the WikiText-103 validation set. The span length follows a Geometric($\mu = 3$) distribution, while the mask ratio varies between 0.1 and 0.9. The sequence length is set to 1024, which aligns with the training setup. (a) Results for the CAR task. (b) Results for the AC task. Tracformer consistently outperforms all baselines.

Table 8: CAR Text infilling performance evaluated on 1BW and Lambada Datasets.

| Mask ranges | MAUVE ($\uparrow$) | | BERT Score ($\uparrow$) | |
|---|---|---|---|---|
| | Tracformer | BART | Tracformer | BART |
| 1BW Dataset | | | | |
| [0.25,0.75] | 0.974 | **0.975** | **0.471** | 0.446 |
| [0.50,1.00] | **0.421** | 0.384 | **0.478** | 0.476 |
| [0.10,0.40] & [0.60,0.90] | **0.992** | 0.979 | **0.377** | 0.334 |
| [0.00,0.40] & [0.50,0.80] | **0.987** | 0.977 | **0.272** | 0.230 |
| [0.00,0.25] & [0.75,1.00] | **0.379** | 0.375 | **0.450** | 0.425 |
| [0.00,0.10] & [0.20,0.50] & [0.70,1.00] | 0.346 | **0.364** | **0.270** | 0.231 |
| Lambada Dataset | | | | |
| [0.25,0.75] | 0.897 | **0.913** | **0.301** | 0.289 |
| [0.50,1.00] | **0.024** | 0.016 | **0.344** | 0.320 |
| [0.10,0.40] & [0.60,0.90] | **0.714** | 0.693 | **0.178** | 0.153 |
| [0.00,0.40] & [0.50,0.80] | 0.804 | **0.839** | **0.077** | 0.056 |
| [0.00,0.25] & [0.75,1.00] | **0.037** | 0.024 | **0.322** | 0.309 |
| [0.00,0.10] & [0.20,0.50] & [0.70,1.00] | **0.023** | 0.014 | **0.068** | 0.041 |

# C. Details of Large-Scale Experiments on WebText/OpenWebText

## C.1. Training Mask Strategy

To scale Tracformer to OpenWebText while balancing both conditional and unconditional generation performance, we adopt a **mixed masking strategy**. This approach integrates contextual autoregressive (CAR) training with an autoregressive (AR) objective by varying the masking strategy probabilistically during training.

At each training iteration, we select from the following masking strategies:

- **Unconditional training (30%)**: No context is provided, and the model learns in a fully autoregressive (AR) manner, simulating traditional language modeling.

- **High-span masking (20%)**: A high masking probability (85%) is applied, forcing the model to rely on limited observed tokens to reconstruct missing content.

- **Moderate-span masking (50%)**: A standard masking probability (50%) is used, following the CAR objective.

The `generate_span_mask` function, as defined in Algorithm 1, is used when applying high- and moderate-span masking strategies to ensure structured token corruption. However, in unconditional training, no span masking is applied, effectively reducing the task to standard autoregressive modeling. This hybrid approach allows Tracformer to retain strong unconditional generation capabilities while improving its ability to generalize to arbitrary conditional queries.

Table 9: CAR Text infilling performance (BLEU-4) evaluated on WikiText103.

| Mask ranges | BLEU-4 ($\uparrow$) | |
| --- | --- | --- |
| | Tracformer | BART |
| [0.25,0.75] | **0.524** | 0.513 |
| [0.50,1.00] | **0.540** | 0.519 |
| [0.10,0.40] & [0.60,0.90] | **0.419** | 0.405 |
| [0.00,0.40] & [0.50,0.80] | **0.339** | 0.325 |
| [0.00,0.25] & [0.75,1.00] | **0.536** | 0.523 |
| [0.00,0.10] & [0.20,0.50] & [0.70,1.00] | **0.337** | 0.322 |

The formalized procedure for the mixed masking strategy is presented in Algorithm 2.

---

**Algorithm 2** Mixed Masking Strategy for OpenWebText Training

---

1: **Input:** Input sequence $\mathbf{x}$ of length $L = 1024$, mask token ID $m_{id}$
2: Sample $r \sim \text{Uniform}(0, 1)$
3: **if** $r < 0.3$ **then**                                                          $\triangleright$ Unconditional training (AR)
4:     $\mathbf{x}_{corrupt} = m_{id}$ (fully masked)
5: **else if** $r < 0.5$ **then**                                                  $\triangleright$ High-span masking (CAR)
6:     $\text{spans}, \text{blank\_ids} = \text{generate\_span\_mask}(L = 1024, p = 0.85, m = 50, \text{distribution type} = \text{Geometric})$
7: **else**                                                                      $\triangleright$ Moderate-span masking (CAR)
8:     $\text{spans}, \text{blank\_ids} = \text{generate\_span\_mask}(L = 1024, p = 0.5, m = 50, \text{distribution type} = \text{Geometric})$
9: **end if**
10: $\mathbf{x}_{corrupt} = \mathbf{x}.\text{clone}()$
11: $\mathbf{x}_{corrupt}[\text{blank\_ids}] = m_{id}$
12: **Return:** $\mathbf{x}_{corrupt}$

---

### C.2. Model and Training Configuration

**Tracformer.** Details of the architecture and optimization procedure for Tracformer are detailed in Table 10.

**SEDD.** We use the SEDD-small model with 169M parameters (including 90M non-embedding parameters) trained on OpenWebText. The model is accessed through HuggingFace: `https://huggingface.co/louaaron/sedd-small`. Following the original paper (Lou et al., 2023), we adopt the log-linear noise schedule and the absorbing mask forward noising process.

**MDLM.** We use the MDLM model with 169M parameters (including 90M non-embedding parameters) trained on OpenWebText. The model is accessed through HuggingFace: `https://huggingface.co/kuleshov-group/mdlm-owt`. Following the original paper (Sahoo et al., 2024), we adopt the log-linear noise schedule.

**GPT.** We use the GPT-2 small model with 124M parameters (including 85M non-embedding parameters). The model is obtained from HuggingFace: `https://huggingface.co/openai-community/gpt2`.

### C.3. Evaluation Metrics

For all conditional generation experiments in Section 6.2, we use sequences of length 128 following prior work (Lou et al., 2023; Han et al., 2023; Gu et al., 2022). Additionally, we exclude EOS tokens when evaluating conditional perplexity since the focus of these tasks is the ability of different models to generate coherent text given prompt texts.

## D. Additional Ablation Studies

To further evaluate the impact of Tracformer's *multi-scope attention*, we conduct an ablation study by replacing it with *full causal attention*. In this modified architecture, the prefix encoder employs *classical dense causal self-attention*, while the suffix encoder uses *reverse causal attention*, where each token attends to all subsequent tokens within the layer. Apart from this modification, the encoder-decoder structure remains unchanged, ensuring a fair comparison.

Table 10: Hyperparameters of Tracformer for the large-scale (OpenWebText) Experiments

| Parameter | Value |
|---|---|
| *Structure Parameters* | |
| Block size | 1024 |
| Number of encoder layers | 10 |
| Number of decoder layers | 10 |
| Max attended tokens ($N_{\text{max}}$) | 32 |
| *Attention Parameters* | |
| Number of attention heads | 9 |
| Embedding dimension ($d$) | 576 |
| *Other Parameters* | |
| Dropout rate | 0.0 |
| *Optimization Parameters* | |
| Initial learning rate | 6e-4 |
| Terminal learning rate | 6e-5 |
| Learning rate schedule | cosine |
| # warmup steps | 2,000 |
| # training steps | 600,000 |
| Batch size | 480 |
| Weight decay | 1e-1 |
| Adam betas | $(0.9, 0.95)$ |
| Gradient clipping maximum norm | 1.0 |

Tables 11 present the AC generation performance under two different masking strategies: *Geometric(50)* with a mean span length of 50, aligned with the training mask length, and *Geometric(10)* with a mean span length of 10, which deviates from the training distribution. The results show that at **low mask ratios** (i.e., when more context is available), both models achieve comparable performance, with the ablation model exhibiting a slight advantage. However, as the **mask ratio increases** and the task becomes more challenging, Tracformer with *multi-scope attention* significantly outperforms the ablation model, demonstrating superior generalization under high-uncertainty conditions.

Another key observation is that replacing multi-scope attention with full attention introduces a notable **efficiency drop**. The ablation model incurs higher computational costs due to the dense attention mechanism, making it less scalable compared to Tracformer. This highlights the practical advantage of Tracformer's sparse *multi-scope self-attention*, which achieves a favorable balance between performance and efficiency.

Finally, it is worth noting that despite replacing multi-scope attention with full attention, the ablation model still **significantly outperforms the previous baseline, BERT-bidir**, which has a similar number of parameters. This underscores the importance of Tracformer's specialized *encoder-decoder design*, beyond just its attention mechanism. The architectural improvements introduced in Tracformer contribute substantially to its ability to handle AC generation tasks, making it a more effective approach compared to conventional NAR models.

## E. Training and Inference Efficiency of Tracformers

**Training Efficiency.** As shown in Section 4.1 and Appendix A, given a Tracformer with $L$ layers, input length $T$, and $N_{\text{max}}$, the computation cost of the encoder is $\mathcal{O}(L \cdot T \cdot N_{\text{max}})$, which is linear w.r.t. $T$. We note that a custom kernel implementation is needed to fully take advantage of this sparse attention and leave it to future work.

According to the decoder attention mask (Eqs. (5) and (6)), the number of tokens required to attend to is exponentially

Table 11: Ablation results for the CAR generation task. *Tracformer w/ full encoder* is an ablation model where the multi-scope encoder in the original Tracformer is replaced with dense self-attention while maintaining the encoder-decoder structure. The table presents results for two span length distributions: Geometric($\mu = 50$) and Geometric($\mu = 10$), evaluated under different mask ratios.

| Model | Mask Ratio | | | | | | | | |
|---|---|---|---|---|---|---|---|---|---|
| | 0.1 | 0.2 | 0.3 | 0.4 | 0.5 | 0.6 | 0.7 | 0.8 | 0.9 |
| | Geometric($\mu = 50$) | | | | | | | | |
| Tracformer | 445.46 | 467.03 | 478.13 | 475.65 | 459.17 | 460.53 | 447.84 | 449.77 | 564.32 |
| Tracformer w/ full encoder | 443.65 | 462.90 | 475.25 | 473.85 | 458.77 | 462.30 | 452.11 | 455.23 | 574.73 |
| | Geometric($\mu = 10$) | | | | | | | | |
| Tracformer | 165.07 | 178.77 | 181.89 | 178.79 | 185.38 | 193.40 | 219.13 | 288.91 | 509.35 |
| Tracformer w/ full encoder | 160.39 | 174.64 | 177.88 | 176.36 | 183.01 | 191.53 | 217.69 | 288.45 | 519.73 |

smaller for initial decoder layers. Suppose $L = \mathcal{O}(\log T)$, then the total computation cost for the decoder is

$$\underbrace{\sum_{l=1}^{L} \mathcal{O}(T \cdot 2^l)}_{\text{Cross attention layers}} + \underbrace{\mathcal{O}(T \cdot L)}_{\text{FFNs}} = \mathcal{O}(T^2 + T \cdot L).$$

This improves upon the cost $\mathcal{O}(T^2 \cdot L)$ in classic decoder-only Transformers.

**Inference Efficiency.** We give an example of using KV-caching techniques in Tracformers designed for CAR generation (i.e., Fig. 4(a)). Given context $\boldsymbol{x}_C$, we first run the suffix encoder for all tokens by feeding the <MASK> token to every token $t \notin C$. We then autoregressively decode the remaining tokens following the order $\boldsymbol{x}_1, \boldsymbol{x}_2, \ldots, \boldsymbol{x}_T$. When decoding $\boldsymbol{x}_t$, we first compute the $(t-1)$-th feature in each layer of the prefix encoder (i.e., $\{\boldsymbol{h}_{t-1}^{\text{enc},l}\}_{l=0}^{L}$). In the attention layers, we use KV caches for preceding tokens to avoid re-computing their features. Then, we compute the $t$-th feature in each layer of the decoder (i.e., $\{\boldsymbol{h}_t^{\text{dec},l}\}_{l=0}^{L}$) to decode $\boldsymbol{x}_t$. This is possible because we have already computed features $\{\boldsymbol{h}_{1:t-1}^{\text{enc},l}\}_{l=0}^{L}$ in the prefix encoder and all features in the suffix encoder. Since there is no self-attention layer in the decoder, KV caching is not needed.

# F. Text Samples from Tracformer

### F.1. Unconditional Generation

The following are randomly selected unconditional text samples generated by a Transformer model with a sequence length of 128:

> in the fishing hole outburst at or before the time that the whale had successfully spawned. Indeed, a relatively close encounter may have emboldened the whale to pull the mylar through the sand and shelled it, as she was able to in this example. Interestingly, the flattening events which resulted in the. Whale cutluring was caused by the thermal spark that resulted from the spin and instability of the high (or low) pressure, thus inducing the ripening state. The more the thermal increase of the high (or low) temperature was offset from the opposite direction along seismic quadrants. Interestingly, the evaporation rates at the ...

> the Antiquarian League of Massachusetts (ASML) works with the Massachusetts League of New Hampshire Bar Associations to enforce the law through nicety and compromise. Unlike the League of Massachusetts Missouri, and its members are independent, their committees are almost entirely autonomous, with sole jurisdiction solely on the matters of labor rights for inmates. They keep offsite their legal animosity to the state, only restricting their manifestations to marking weeks in advance.And, in an unusual turn, they hold committees to run stretchin, direct sensitivity exercises of a ceremonies-like kind. These bundles of letters in the verse, in order to mark the time during which a batch ...

The 90-day player initiative on June 2 also launched across all seven parks and recreational areas in the National Park system in Central Sierras. There, children along with the government hamstrung the operation, an action hoped to be a long-term solution to the larger problem of ecological loss along the Central Sierras. The USDA estimated that 20 275-acre sites on the Hudson River basin are almost colliding, forming a "very difficult interplay of life and conflict." They include the East Bay Schuylkill River (230 versus the 81-acre Wyoming Standard Ditch), the ...

was directed against the convict institution. Augustine said the parish sincerely believed the practice was against the 100 U.S. Statute. In fact if that is the practice, we would not call it a crime that was committed in Phillippines – we were talking about out of state institutions. And we don't do that. So if the confession was confessionally formed it is in the whole, full church confessional as we described it."\n\nAgain, the most surprising day of the probation hearing was the fact the single author testified the confession was genuine.\n\nThe principal tried to inscribe the patriarch, ...

I still see themes that clouds up uncertainty about the work of first Y.\n\nInterestingly, note how Elsa feels an inability to see how far to go on her journey in those citymarish dark castles. Darkness isn't the right word for her solution to the riddle – the life of an urban teen slowly dissolving behind the clouds, every shadow suddenly overwhelmed or broken, only to be replaced by a more believable and understandable life. The crowding of darkness adds to the disorder. Of course she doesn't expect Snow to go though, and Elsa will have to find her own answers as to how far ...

TalkTalk Studios production. "We're stellar!"!\n\nAt the time of the movie's release, there were quite a few references to a 5.0 rating and 60% top ballerinas, plus amusing examples of mixed reviews and negative news coverage. "We really appealed to the social media revolution because I think that's really got to get people excited and care about their content: whether they like it or not," producer Jill Mullin has explained.\n\nA downhill slope has been one of the biggest tickets for Sony's likes-on-rumours

## F.2. Conditional Generation

The following are randomly selected conditional text samples generated by a Transformer model with a sequence length of 128. The prompt texts are bolded and in blue:

**track-two diplomacy between U.S. and Soviet officials and nuclear scientists, which helped lead to the Comprehensive Test Ban Treaty. MacArthur grantees also helped develop** real alternatives to the IAEA, such as providing uranium to New Zealand. The centerpiece was the launch of a satellite called WUSA.\n\nFinances of the IAEA declined. The international community supports the cessation of proliferation, nuclear nonproliferation, and proliferation reduction, and it has improved the coordination of **security policy on issues like nonproliferation and arms control.But more recently, they have borne the brunt of its economic decline, enduring lower wages**

**and to ascertain its peaceful means of production.Mugabe has ruled Zimbabwe since independence from Britain in 1980 but faced an unprecedented challenge in Saturday's elections because of** the unrest across his country.\n\nFrom newly jailed opposition leader Eldion Oliverine for the Republic of Zimbabwe to the conservative populist Robert Vibert for the ruling party, the verdict could mark a "new beginning" for Zimbabwe, which had once been its own country.\n\nThe verdict applies to **brain cancer, O. Wayne Corley, senior shareholder in the McNair Law Firm, announced.Another question that seems never to have been raised is that**

Powerton Historic District. Nearly 1 million square feet may be taken care of by the Virginia Historical Commission, say officials, but the built-design preferred by the Lee administration would have led to **up to six months.Thousands of jobs are threatened, and towns will lose part of their economic base.OK, so** today is time for some Aliens Interviews: Episode 5 – Gruff Ed Beeton interview.\n\nAs you might expect but a few comments are here:\n\nTerm Ear Ham Mick **Deputy Assistant to the President, Mr. Ballentine was Special Assistant to the President for Legislative Affairs, where he focused on energy**

full-time practice for the Dynamo with one of their available options to make his debut from the bench is looking likely to be the more familiar striker. The way players are used to it is not the "main factor," Roden said.If there are rights, it is too little.He had been in San Antonio since the winter of 2010 as a civil servant with a pension to return to Brazil.\n\n"Brazil is the kind of economy that opened up variations in economic conditions and strengthened the key refinery centers," said Jim Rouiller, meteorologist with private weather forecaster Planalytics.Headquartered in Basking

at a wavelength of 13 cm show no evidence for water ice (Fig. 16(j)). The middle crust of the pixel is depressurised by around 0.3 Å before a crystalline bath and column. The software eliminates any liquid after residual ice, down to the image resolution of 10 m per pixel.\n On November 15, 2008, a 34-kg, 5.2-pound airborne "mysterious satellite" appears on high resolution radar screen at altimeter branches at the Pangaea solar array observatory on Oak Flat, California, with a radar altimeter, video imaging system, and a mass spect

ramuros, were destroyed but after the war, reconstruction took place.\n In 1948, President Elpidio Quiroga set in motion a program that could reshape Philippine history. The concept was called "The Bilateral Relations with Manila, created in 1938 by former President Manuel L. Quezon, which was named after him. The move ended any implementation first created by then-President Marcos.\n\nIn 1945, the Bilateral Relations obscure evolved into inter-provincial relations. Manila and Bilateral Relations were also linked to the Bilateral Relations.\n\nThe Bilateral relations were based on strategic, tactical

is difficult to make sense of. Most researchers think that terrestrial vertebrae are made of tendon, but there are several proposed hypotheses.1 Two proposed vertebrae, which sometimes don't such as Sclerothorax and Eryops that may have been at least partly terrestrial also have long neural spines on top of their vertebrae that would have stabilized the spine and required reattachment to form a non-spine rest part.3 Another speculation involving the placement of vertebrae in a notch below the flexic muscles on the back of the spine would seem a more plausible explanation.3 Another proposed vertebrae

## G. Proof of Conditional NELBO for MDLM

In this section, we generalize the unconditional negative evidence lower-bound (NELBO) derivation in Sahoo et al. (2024) to the conditional case. The resultant formula is then used to compute the conditional perplexity of baseline discrete diffusion models in various settings. First, we derive the expression for conditional NELBO. We then examine and simplify the various terms within the conditional NELBO. Finally, we present the final expression for the conditional NELBO.

This proof builds upon and extends the results presented in Sahoo et al. (2024). Equations from the original paper are frequently referenced, and readers are encouraged to refer to the original paper for a deeper understanding of the foundational concepts and equations discussed.

### G.1. Derivation of the Conditional NELBO Expression

For a sequence of $L$ tokens, let $F = \{1, 2, \ldots, L\}$ denote the set of all indices in the sequence. Let $S \subset F$ be the set containing all and only the indices of the given tokens. In this proof, we denote the entire sequence as $\boldsymbol{x}^F$ (equivalent to $\boldsymbol{x}_{1:L}$ in Sahoo et al. (2024)), the given tokens as $\boldsymbol{x}^S$, and the unknown tokens as $\boldsymbol{x}^{F-S}$. Following the definitions in Sahoo et al. (2024), we define $s(i) = (i-1)/T$ and $t(i) = i/T$, and omit the $i$ from $t(i)$ and $s(i)$.

Using the aforementioned notations, we can express the conditional negative log-likelihood as $-\log p_\theta(\boldsymbol{x}^{F-S}|\boldsymbol{x}^S)$. Now, we step into the details of deriving the conditional NELBO:

$$-\log p_\theta(\boldsymbol{x}^{F-S}|\boldsymbol{x}^S),$$
$$= -\log \int p_\theta(\boldsymbol{x}^{F-S}, \boldsymbol{z}^F_{t(0)}, \ldots, \boldsymbol{z}^F_{t(T)}|\boldsymbol{x}^S) d\boldsymbol{z}^F_{t(0)} \ldots d\boldsymbol{z}^F_{t(T)},$$
$$= -\log \int q(\boldsymbol{z}^F_{t(0)}, \ldots, \boldsymbol{z}^F_{t(T)}|\boldsymbol{x}^F) \frac{p_\theta(\boldsymbol{x}^{F-S}, \boldsymbol{z}^F_{t(0)}, \ldots, \boldsymbol{z}^F_{t(T)}|\boldsymbol{x}^S)}{q(\boldsymbol{z}^F_{t(0)}, \ldots, \boldsymbol{z}^F_{t(T)}|\boldsymbol{x}^F)} d\boldsymbol{z}^F_{t(0)} \ldots d\boldsymbol{z}^F_{t(T)},$$
$$\leq -\int q(\boldsymbol{z}^F_{t(0)}, \ldots, \boldsymbol{z}^F_{t(T)}|\boldsymbol{x}^F) \log \frac{p_\theta(\boldsymbol{x}^{F-S}, \boldsymbol{z}^F_{t(0)}, \ldots, \boldsymbol{z}^F_{t(T)}|\boldsymbol{x}^S)}{q(\boldsymbol{z}^F_{t(0)}, \ldots, \boldsymbol{z}^F_{t(T)}|\boldsymbol{x}^F)} d\boldsymbol{z}^F_{t(0)} \ldots d\boldsymbol{z}^F_{t(T)},$$

$$= -\mathbb{E}_{\boldsymbol{z}^F \sim q(\cdot|\boldsymbol{x}^F)} \left[ \log \frac{p_\theta(\boldsymbol{x}^{F-S}, \boldsymbol{z}^F_{t(0)}, \dots, \boldsymbol{z}^F_{t(T)}|\boldsymbol{x}^S)}{q(\boldsymbol{z}^F_{t(0)}, \dots, \boldsymbol{z}^F_{t(T)}|\boldsymbol{x}^F)} \right], \tag{9}$$

$$= -\mathbb{E}_{\boldsymbol{z}^F} \left[ \log \frac{p_\theta(\boldsymbol{x}^{F-S}|\boldsymbol{z}^F_{t(0)}, \boldsymbol{x}^S) p_\theta(\boldsymbol{z}^F_{t(T)}|\boldsymbol{x}^S) \prod_{i=1}^T p_\theta(\boldsymbol{z}^F_s|\boldsymbol{z}^F_t, \boldsymbol{x}^S)}{q(\boldsymbol{z}^F_{t(0)}|\boldsymbol{x}^F) \prod_{i=1}^T q(\boldsymbol{z}^F_t|\boldsymbol{z}^F_s)} \right] \quad \text{(note that } s \text{ (resp. } t) \text{ denotes } s(i) \text{ (resp. } t(i))),$$

$$= \mathbb{E}_{\boldsymbol{z}^F} \left[ -\log p_\theta(\boldsymbol{z}^F_{t(T)}|\boldsymbol{x}^S) + \sum_{i=1}^T \log \frac{q(\boldsymbol{z}^F_t|\boldsymbol{z}^F_s)}{p_\theta(\boldsymbol{z}^F_s|\boldsymbol{z}^F_t, \boldsymbol{x}^S)} + \log \frac{q(\boldsymbol{z}^F_{t(0)}|\boldsymbol{x}^F)}{p_\theta(\boldsymbol{x}^{F-S}|\boldsymbol{z}^F_{t(0)}, \boldsymbol{x}^S)} \right],$$

$$= \mathbb{E}_{\boldsymbol{z}^F} \left[ -\log p_\theta(\boldsymbol{z}^F_{t(T)}|\boldsymbol{x}^S) + \sum_{i=1}^T \log \frac{q(\boldsymbol{z}^F_s|\boldsymbol{z}^F_t, \boldsymbol{x}^F)}{p_\theta(\boldsymbol{z}^F_s|\boldsymbol{z}^F_t, \boldsymbol{x}^S)} \right.$$
$$\left. + \sum_{i=1}^T \log \frac{q(\boldsymbol{z}^F_t|\boldsymbol{x}^F)}{q(\boldsymbol{z}^F_s|\boldsymbol{x}^F)} + \log \frac{q(\boldsymbol{z}^F_{t(0)}|\boldsymbol{x}^F)}{p_\theta(\boldsymbol{x}^{F-S}|\boldsymbol{z}^F_{t(0)}, \boldsymbol{x}^S)} \right], \tag{10}$$

$$= \mathbb{E}_{\boldsymbol{z}^F} \left[ -\log p_\theta(\boldsymbol{z}^F_{t(T)}|\boldsymbol{x}^S) + \sum_{i=1}^T \log \frac{q(\boldsymbol{z}^F_s|\boldsymbol{z}^F_t, \boldsymbol{x}^F)}{p_\theta(\boldsymbol{z}^F_s|\boldsymbol{z}^F_t, \boldsymbol{x}^S)} + \log \frac{q(\boldsymbol{z}^F_{t(T)}|\boldsymbol{x}^F)}{q(\boldsymbol{z}^F_{t(0)}|\boldsymbol{x}^F)} + \log \frac{q(\boldsymbol{z}^F_{t(0)}|\boldsymbol{x}^F)}{p_\theta(\boldsymbol{x}^{F-S}|\boldsymbol{z}^F_{t(0)}, \boldsymbol{x}^S)} \right],$$

$$= \mathbb{E}_{\boldsymbol{z}^F} \left[ \log \frac{q(\boldsymbol{z}^F_{t(T)}|\boldsymbol{x}^F)}{p_\theta(\boldsymbol{z}^F_{t(T)}|\boldsymbol{x}^S)} + \sum_{i=1}^T \log \frac{q(\boldsymbol{z}^F_s|\boldsymbol{z}^F_t, \boldsymbol{x}^F)}{p_\theta(\boldsymbol{z}^F_s|\boldsymbol{z}^F_t, \boldsymbol{x}^S)} - \log p_\theta(\boldsymbol{x}^{F-S}|\boldsymbol{z}^F_{t(0)}, \boldsymbol{x}^S) \right],$$

$$= \mathbb{E}_{\boldsymbol{z}^F} \left[ \underbrace{-\log p_\theta(\boldsymbol{x}^{F-S}|\boldsymbol{z}^F_{t(0)}, \boldsymbol{x}^S)}_{\mathcal{L}_{\text{recons}}} + \underbrace{\sum_{i=1}^T D_{\text{KL}}[q(\boldsymbol{z}^F_s|\boldsymbol{z}^F_t, \boldsymbol{x}^F) \| p_\theta(\boldsymbol{z}^F_s|\boldsymbol{z}^F_t, \boldsymbol{x}^S)]}_{\mathcal{L}_{\text{diffusion}}} \right]$$
$$+ \underbrace{D_{\text{KL}}[q(\boldsymbol{z}^F_{t(T)}|\boldsymbol{x}^F) \| p_\theta(\boldsymbol{z}^F_{t(T)}|\boldsymbol{x}^S)]}_{\mathcal{L}_{\text{prior}}}. \tag{11}$$

In Equation (9), $\boldsymbol{z}^F$ is used to denote $[\boldsymbol{z}^F_{t(0)}, \dots, \boldsymbol{z}^F_{t(T)}]$ for simplicity. When deriving Equation (10) from its previous step, we make use of the following factorization:

$$q(\boldsymbol{z}^F_t|\boldsymbol{z}^F_s) = q(\boldsymbol{z}^F_t|\boldsymbol{z}^F_s, \boldsymbol{x}^F) = \frac{q(\boldsymbol{z}^F_s|\boldsymbol{z}^F_t, \boldsymbol{x}^F) q(\boldsymbol{z}^F_t|\boldsymbol{x}^F)}{q(\boldsymbol{z}^F_s|\boldsymbol{x}^F)}.$$

## G.2. Simplification of Conditional NELBO Terms

As assumed in Section 3.5 of Sahoo et al. (2024), the forward noising process is applied independently across a sequence, and the denoising process factorizes independently across tokens. These two assumptions can be translated to the following equations:

$$q(\boldsymbol{z}^F_t|\boldsymbol{z}^F_s, \boldsymbol{x}^F) = \prod_{\ell=1}^L q(\boldsymbol{z}^\ell_t|\boldsymbol{z}^\ell_s, \boldsymbol{x}^\ell) \tag{12}$$

$$p_\theta(\boldsymbol{z}^F_s|\boldsymbol{z}^F_t) = \prod_{\ell=1}^L p_\theta(\boldsymbol{z}^\ell_s|\boldsymbol{z}^F_t) \tag{13}$$

As defined in Equation (7) of Sahoo et al. (2024), for $\forall \ell \in F$,

$$p_\theta(\boldsymbol{z}^\ell_s|\boldsymbol{z}^F_t) = q(\boldsymbol{z}^\ell_s|\boldsymbol{z}^\ell_t, \boldsymbol{x}^\ell = \boldsymbol{x}^\ell_\theta(\boldsymbol{z}^F_t, t)) \tag{14}$$

Note that the original definition was for the specific case where only a single token is noised and denoised. However, since we are addressing the noising and denoising of entire token sequences, we extend the definition to accommodate sequences of tokens.

Additionally, we define:

$$p_\theta(\boldsymbol{z}_s^\ell | \boldsymbol{z}_t^F, \boldsymbol{x}^S) = q(\boldsymbol{z}_s^\ell | \boldsymbol{z}_t^\ell, \boldsymbol{x}^\ell = \boldsymbol{x}_\theta^\ell([\boldsymbol{z}_t^{F-S}, \boldsymbol{x}^S], t)) \tag{15}$$

where $[\boldsymbol{z}_t^{F-S}, \boldsymbol{x}^S]$ is a shorthand for $\boldsymbol{z}_t^F$ with $\boldsymbol{z}_t^\ell$ replaced by $\boldsymbol{x}^\ell$, for $\forall \ell \in S$. In practice, this is easily done by replacing the tokens with indices in $S$ with the corresponding given tokens, while leaving the remaining tokens unchanged before feeding the sequence into the model $\boldsymbol{x}_\theta$.

According to "Carry Over Masking" in the original paper, the model $\boldsymbol{x}_\theta$ has the property that $\boldsymbol{x}_\theta^\ell(\boldsymbol{z}_t^F, t) = \boldsymbol{z}_t^\ell$ if $\boldsymbol{z}_t^\ell$ is not masked. Therefore, for $\forall \ell \in S$,

$$\begin{aligned}
p_\theta(\boldsymbol{z}_s^\ell | \boldsymbol{z}_t^F, \boldsymbol{x}^S) &= q(\boldsymbol{z}_s^\ell | \boldsymbol{z}_t^\ell, \boldsymbol{x}^\ell = \boldsymbol{x}_\theta^\ell([\boldsymbol{z}_t^{F-S}, \boldsymbol{x}^S], t)) \\
&= q(\boldsymbol{z}_s^\ell | \boldsymbol{z}_t^\ell, \boldsymbol{x}^\ell)
\end{aligned} \tag{16}$$

Note that the symbol the $\boldsymbol{x}^\ell$ is reused: the $\boldsymbol{x}^\ell$ in the first line represents a random variable, while the $\boldsymbol{x}^\ell$ in the second line is the $\ell$-th token of the ground truth sequence, which is a given one-hot vector.

As an extension to (13), we further assume that:

$$p_\theta(\boldsymbol{z}_s^F | \boldsymbol{z}_t^F, \boldsymbol{x}^S) = \prod_{\ell=1}^L p_\theta(\boldsymbol{z}_s^\ell | \boldsymbol{z}_t^F, \boldsymbol{x}^S) \tag{17}$$

Using these equations and the continuation of timesteps (i.e. $T \to \infty$), we can further simplify the terms in (11).

### G.2.1. DIFFUSION LOSS

Following Section A.2.3 of the original paper, let $\mathcal{L}_T = \mathbb{E}_{t \in \{\frac{1}{T}, \frac{2}{T}, \dots, 1\}} \mathbb{E}_{q(\boldsymbol{z}_t^F | \boldsymbol{x}^F)} T D_{\mathrm{KL}}[q(\boldsymbol{z}_s^F | \boldsymbol{z}_t^F, \boldsymbol{x}^F) \| p_\theta(\boldsymbol{z}_s^F | \boldsymbol{z}_t^F, \boldsymbol{x}^S)]$ denote the diffusion loss. We can simplify it with the following steps:

$$\begin{aligned}
\mathcal{L}_T &= \mathbb{E}_{t \in \{\frac{1}{T}, \frac{2}{T}, \dots, 1\}} \mathbb{E}_{q(\boldsymbol{z}_t^F | \boldsymbol{x}^F)} T \sum_{\ell=1}^L D_{\mathrm{KL}}[q(\boldsymbol{z}_s^\ell | \boldsymbol{z}_t^\ell, \boldsymbol{x}^\ell) \| p_\theta(\boldsymbol{z}_s^\ell | \boldsymbol{z}_t^F, \boldsymbol{x}^S)], && \text{Using (12) and (17)} \\
&= \mathbb{E}_{t \in \{\frac{1}{T}, \frac{2}{T}, \dots, 1\}} \mathbb{E}_{q(\boldsymbol{z}_t^F | \boldsymbol{x}^F)} T \sum_{\ell \in F-S} D_{\mathrm{KL}}[q(\boldsymbol{z}_s^\ell | \boldsymbol{z}_t^\ell, \boldsymbol{x}^\ell) \| p_\theta(\boldsymbol{z}_s^\ell | \boldsymbol{z}_t^F, \boldsymbol{x}^S)], && \text{Using (16)} \\
&= \sum_{\ell \in F-S} \mathbb{E}_{t \in \{\frac{1}{T}, \frac{2}{T}, \dots, 1\}} \mathbb{E}_{q(\boldsymbol{z}_t^F | \boldsymbol{x}^F)} \left[ T \cdot D_{\mathrm{KL}}[q(\boldsymbol{z}_s^\ell | \boldsymbol{z}_t^\ell, \boldsymbol{x}^\ell) \| p_\theta(\boldsymbol{z}_s^\ell | \boldsymbol{z}_t^F, \boldsymbol{x}^S)] \right],
\end{aligned}$$

which is essentially the sum of the losses for each unknown token.

In the appendix of Sahoo et al. (2024), the derivation is focused on the case where only a single token is involved. But since we decomposed the diffusion loss of the sequence into the sum of single-token losses, now we are able to use the results from the original paper, as long as we can address the difference between $p_\theta(\boldsymbol{z}_s^\ell | \boldsymbol{z}_t^F, \boldsymbol{x}^S)$ in this derivation and $p_\theta(\boldsymbol{z}_s | \boldsymbol{z}_t)$ in the original paper.

According to Equation (15) from the original paper, we can derive from Equation (15) that:

$$p_\theta(\boldsymbol{z}_s^\ell | \boldsymbol{z}_t^F, \boldsymbol{x}^S) = q(\boldsymbol{z}_s^\ell | \boldsymbol{z}_t^\ell, \boldsymbol{x}^\ell = \boldsymbol{x}_\theta^\ell([\boldsymbol{z}_t^{F-S}, \boldsymbol{x}^S], t)) = \mathrm{Cat}\left( \boldsymbol{z}_s^\ell; \frac{Q_{t|s} \boldsymbol{z}_t^\ell \odot Q_s^\top \boldsymbol{x}_\theta^\ell([\boldsymbol{z}_t^{F-S}, \boldsymbol{x}^S], t))}{\boldsymbol{z}_t^{\ell\top} Q_t^\top \boldsymbol{x}_\theta^\ell([\boldsymbol{z}_t^{F-S}, \boldsymbol{x}^S], t))} \right).$$

Thus, the KL divergence term in the diffusion loss can be expressed as:

$$D_{\mathrm{KL}}[q(\boldsymbol{z}_s^\ell | \boldsymbol{z}_t^\ell, \boldsymbol{x}^\ell) \| p_\theta(\boldsymbol{z}_s^\ell | \boldsymbol{z}_t^F, \boldsymbol{x}^S)] = D_{\mathrm{KL}}\left[ q(\boldsymbol{z}_s^\ell | \boldsymbol{z}_t^\ell, \boldsymbol{x}^\ell) \| \mathrm{Cat}\left( \boldsymbol{z}_s^\ell; \frac{Q_{t|s} \boldsymbol{z}_t^\ell \odot Q_s^\top \boldsymbol{x}_\theta^\ell([\boldsymbol{z}_t^{F-S}, \boldsymbol{x}^S], t))}{\boldsymbol{z}_t^{\ell\top} Q_t^\top \boldsymbol{x}_\theta^\ell([\boldsymbol{z}_t^{F-S}, \boldsymbol{x}^S], t))} \right) \right], \tag{18}$$

while for the single-token case as in the original paper, the KL divergence term $D_{\mathrm{KL}}[q(\boldsymbol{z}_s^\ell | \boldsymbol{z}_t^\ell, \boldsymbol{x}^\ell) \| p_\theta(\boldsymbol{z}_s^\ell | \boldsymbol{z}_t^\ell)]$ is:

$$D_{\mathrm{KL}}[q(\boldsymbol{z}_s^\ell | \boldsymbol{z}_t^\ell, \boldsymbol{x}^\ell) \| p_\theta(\boldsymbol{z}_s^\ell | \boldsymbol{z}_t^\ell)] = D_{\mathrm{KL}}\left[ q(\boldsymbol{z}_s^\ell | \boldsymbol{z}_t^\ell, \boldsymbol{x}^\ell) \| \mathrm{Cat}\left( \boldsymbol{z}_s; \frac{Q_{t|s} \boldsymbol{z}_t^\ell \odot Q_s^\top \boldsymbol{x}_\theta^\ell(\boldsymbol{z}_t^\ell, t))}{\boldsymbol{z}_t^{\ell\top} Q_t^\top \boldsymbol{x}_\theta^\ell(\boldsymbol{z}_t^\ell, t))} \right) \right]. \tag{19}$$

In Equation (19), the index $\ell \in F - S$ is added for notational consistency between Equation (18) and Equation (19). As we can see, the sole difference is that in Equation (19), $x_\theta^\ell([z_t^{F-S}, x^S], t)$ is substituted with $x_\theta^\ell(z_t^\ell, t)$. Consequently, as long as the former shares the same properties as the latter, the results derived for the single-token case still applies.

When simplifying the diffusion loss, only 3 properties are required of $x_\theta^\ell(z_t^\ell, t)$ in the original paper:

1. The output of $x_\theta^\ell(z_t^\ell, t)$ is a probability distribution over all categories, i.e. $\langle \mathbf{1}, x_\theta^\ell(z_t^\ell, t) \rangle = 1$.

2. "Zero Masking Probabilities" i.e. $\langle x_\theta^\ell(z_t^\ell, t), \mathbf{m} \rangle = 0$.

3. "Carry Over Unmasking" i.e. $x_\theta^\ell(z_t^\ell, t) = x^\ell$ when $z_t^\ell = x^\ell$.

which all happens to be properties that MDLM models share, and therefore also holds for $x_\theta^\ell([z_t^{F-S}, x^S], t)$. Thus, by substituting $x_\theta(z_t, t)$ with $x_\theta^\ell([z_t^{F-S}, x^S], t)$, we can apply the results from the single-token case and derive the following equation from Eqn. 42 of the original paper:

$$\mathcal{L}_T = \sum_{\ell \in F-S} \mathbb{E}_{t \in \{\frac{1}{T}, \frac{2}{T}, \dots, 1\}} \mathbb{E}_{q(z_t^F | x^F)} T \left[ \frac{\alpha_t - \alpha_s}{1 - \alpha_t} \log \langle x_\theta^\ell([z_t^{F-S}, x^S], t), x^\ell \rangle \right]. \tag{20}$$

### G.2.2. PRIOR LOSS

According to Equations (37) and (38) in the original paper, we can easily derive that:

$$q(z_{t(T)}^F = \mathbf{M} | x^F) = 1,$$
$$p_\theta(z_{t(T)}^F = \mathbf{M} | x^S) = 1,$$

where $\mathbf{M} = [\mathbf{m}, \mathbf{m}, \dots, \mathbf{m}] \in \mathbb{R}^{K \times L}$, and $K$ is the number of categories.

Thus, $q(z_{t(T)}^F | x^F)$ and $p_\theta(z_{t(T)}^F | x^S)$ are identical distributions where $z_{t(T)}^F$ has a probability of 1 of being an entire sequence of masked tokens, resulting in a prior loss of 0:

$$\mathcal{L}_{\text{prior}} = D_{\text{KL}}[q(z_{t(T)}^F | x^F) \| p_\theta(z_{t(T)}^F | x^S)] = 0. \tag{21}$$

### G.2.3. CONTINUOUS TIME

For the continuous-time case, similar to Equation (44) from the original paper, we can derive:

$$\mathcal{L}_{\text{diffusion}}^\infty = \lim_{T \to \infty} \mathcal{L}_T = \sum_{\ell \in F-S} \mathbb{E}_{t \sim \mathcal{U}[0,1], q(z_t^F | x^\ell)} \left[ \frac{\alpha_t'}{1 - \alpha_t} \log \langle x_\theta^\ell([z_t^{F-S}, x^S], t), x^\ell \rangle \right]. \tag{22}$$

According to Equation (45) from the original paper, under continuous time, we also have $z_{t(0)}^\ell = x^\ell$. Thus we have the following:

$$
\begin{aligned}
\mathcal{L}_{\text{recons}} &= \mathbb{E}_{q(z_{t(0)}^F | x^F)}[-\log p_\theta(x^{F-S} | z_{t(0)}^F, x^S)], \\
&= \mathbb{E}_{q(z_{t(0)}^F | x^F)}[-\log p_\theta(x^{F-S} | z_{t(0)}^F = x^F, x^S)], && \text{Using } z_{t(0)}^\ell = x^\ell \\
&= \mathbb{E}_{q(z_{t(0)}^F | x^F)}[-\log \prod_{\ell \in F-S} \langle x_\theta^\ell([z_{t(0)}^{F-S} = x^{F-S}, x^S], t(0)), x^\ell \rangle], \\
&= \mathbb{E}_{q(z_{t(0)}^F | x^F)}[-\log \prod_{\ell \in F-S} \langle x^\ell, x^\ell \rangle], && (23) \\
&= \mathbb{E}_{q(z_{t(0)}^F | x^F)}[-\log \prod_{\ell \in F-S} 1], \\
&= 0, && (24)
\end{aligned}
$$

where deriving Equation (23) from its previous step makes use of the "Carry Over Masking" property, i.e., $x_\theta^\ell([z_t^{F-S} = x^{F-S}, x^S], t) = x^\ell$.

### G.2.4. CONDITIONAL NELBO

Finally, using Equations (21), (22) and (24), we arrive at the final expression for conditional NELBO:

$$
\mathcal{L}_{\text{NELBO}} = \mathcal{L}_{\text{recons}} + \mathcal{L}_{\text{diffusion}}^{\infty} + \mathcal{L}_{\text{prior}},
$$
$$
= \sum_{\ell \in F-S} \mathbb{E}_{q,t} \left[ \frac{\alpha_t'}{1 - \alpha_t} \log \langle \boldsymbol{x}_\theta^\ell([\boldsymbol{z}_t^{F-S}, \boldsymbol{x}^S], t), \boldsymbol{x}^\ell \rangle \right].
$$

