# OpenReview forum: "Tractable Transformers for Flexible Conditional Generation"
_ICML.cc/2025/Conference — ICML 2025 poster_

### Official Review · Reviewer_7ULg · 2025-03-03

**Overall Recommendation:** 3

**Summary:**

This paper introduces Tracformer, a Transformer-based generative model designed for flexible and robust conditional generation tasks. Tracformer incorporates a sparse multi-scope attention mechanism to capture both local and global contextual information efficiently. Empirical results demonstrate that Tracformer outperforms existing NAR models like SEDD and MDLM in both conditional and unconditional generation tasks, showcasing its potential as a scalable and versatile generative model.

**Claims And Evidence:**

Yes

**Essential References Not Discussed:**

None

**Experimental Designs Or Analyses:**

Yes

**Methods And Evaluation Criteria:**

Yes

**Other Comments Or Suggestions:**

None

**Other Strengths And Weaknesses:**

Strengths:
1) The motivation for the study is clearly introduced. Non-autoregressive (NAR) models have shown superior performance in unconditional generation compared to autoregressive (AR) models of similar sizes. However, a significant challenge with NAR models is their difficulty in generalizing to conditional probability queries that were not seen during training. In response to this issue, the authors propose the Tractable Transformer (Tracformer), a NAR model designed specifically for flexible conditional generation tasks.
2) The experimental results demonstrate that the proposed Tracformer model outperforms several established models, including BERT, BART, SEDD, and MDLM, in terms of conditional generation performance.

Weaknesses:
1) Some abbreviations are repeatedly defined throughout the paper, which leads to unnecessary redundancy. For example, terms like non-autoregressive (NAR) and feed-forward neural network (FFN) are defined multiple times.
2) The contributions of the paper would be more clearly understood if they were summarized at the end of the Introduction section. This would help provide the reader with a clearer context and set expectations for the rest of the paper.
3) In the experimental section, the authors compare their proposed Tracformer model only with BERT and BART. However, more recent and powerful models, such as GPT, should be included in the comparisons presented in Figures 5 and 6.
4) The results for zero-shot unconditional perplexity in Table 4 show that Tracformer outperforms the other baselines on only the 1BW dataset. In contrast, MDLM demonstrates superior performance on three datasets, while Tracformer only outperforms baselines on one.
5) The Background section could be moved to the appendix. Sequence modeling and Transformer models are well-established concepts and are common knowledge for researchers and readers familiar with this field. Additionally, more detailed experiments and comparisons with recent models, such as GPT, should be conducted and presented in the experimental section. A deeper analysis comparing the proposed model against more state-of-the-art architectures would provide a clearer picture of its strengths and weaknesses.

**Questions For Authors:**

See above.

**Relation To Broader Scientific Literature:**

Fair

**Theoretical Claims:**

Yes

---

> ### Author Rebuttal · Authors · 2025-03-31
>
> We thank the reviewer for their constructive feedback and for recognizing the potential of Tracformers in conditional generation tasks.
>
> > Some abbreviations are repeatedly defined throughout the paper, which leads to unnecessary redundancy.
>
> We appreciate the reviewer’s feedback and will revise the paper to ensure that abbreviations are defined only once.
>
> > The contributions of the paper would be more clearly understood if they were summarized at the end of the Introduction section.
>
> We thank the reviewer for the suggestion. In summary, our contributions are two-fold: (i) we identified the problem that existing NAR models suffer from severe performance degradation in conditional generation, despite having strong unconditional generation performance; (ii) we propose Tracformer, a novel Transformer-based architecture specially designed to improve conditional generation and generalize to conditional queries unseen during training. In the next version of the paper, we will add this summary at the end of the Introduction section to enhance clarity.
>
> > In the experimental section, the authors compare their proposed Tracformer model only with BERT and BART. However, more recent and powerful models, such as GPT, should be included in the comparisons presented in Figures 5 and 6.
>
> While autoregressive models like GPT are indeed very powerful, they cannot be used in the contextual AR (CAR) and arbitrary-context (AC) generation experiments in Figures 5 and 6. This is because GPT models can only condition on prefix prompts, whereas our experiments require conditioning on context that is scattered throughout the sequence.
>
> In our experiments, we included GPT-2 as a baseline in Table 4, as it has a similar number of parameters compared to our model and other baselines. In this comparison, all models were trained on the WebText dataset (or its open-sourced version) and evaluated for their zero-shot unconditional perplexity across multiple datasets.
>
> > The results for zero-shot unconditional perplexity in Table 4 show that Tracformer outperforms the other baselines on only the 1BW dataset.
>
> We acknowledge that the zero-shot unconditional perplexity of Tracformer is less favorable compared to MDLM. However, the primary focus of this paper is to highlight the limitation of existing NAR models, including MDLM, which achieve strong unconditional generation performance but struggle with conditional generation. While Tracformer exhibits slightly worse unconditional generation performance, it significantly outperforms MDLM in multiple conditional generation tasks (as shown in Section 6.2). Since conditional generation is often more critical for downstream applications, we believe this aspect is more important.
>
> Another potential factor influencing the results is that the Tracformer model used in our experiments has fewer parameters (109M vs. 169M) and was trained on fewer tokens (295B vs. 524B). This is summarized in Table 3. In future work, we plan to scale up Tracformers and explore different training objectives, such as the diffusion objective used by MDLM, to further evaluate their scalability and versatility.
>
> > The Background section could be moved to the appendix.
>
> We thank the reviewer for the helpful suggestion. In the next version of the paper, we will move most of the background of sequence modeling and Transformers to the appendix, retaining only the essential parts needed to introduce basic notations.
>
> > …more detailed experiments and comparisons with recent models, such as GPT, should be conducted and presented in the experimental section. A deeper analysis comparing the proposed model against more state-of-the-art architectures would provide a clearer picture of its strengths and weaknesses.
>
> We thank the reviewer for the suggestion. In Section 6.2, we compared Tracformer against state-of-the-art autoregressive models (e.g., GPT-2) as well as non-autoregressive models (e.g., SEDD and MDLM). For unconditional generation tasks, we included both autoregressive and non-autoregressive baselines. For conditional generation tasks, only non-autoregressive models were included since autoregressive models cannot condition on suffix contexts, making a direct comparison infeasible. In the next version of the paper, we will add the above discussion to the experiment section.

---

### Official Review · Reviewer_QzDn · 2025-03-14

**Overall Recommendation:** 3

**Summary:**

This paper explores why non-autoregressive (NAR) generative models often underperform in conditional tasks, despite strong unconditional performance. The authors introduce Tractable Transformers (TracFormer), which factorize conditional queries to handle partial inputs flexibly while leveraging both local and global context. Their experiments show that TracFormer achieves robust conditional generation, surpassing diffusion- and autoregressive-based baselines.

## update after rebuttal
The authors have provided a thorough rebuttal and addressed most of my concerns. I continue to believe that my original score was an accurate evaluation.

**Claims And Evidence:**

Yes, the observations are clearly articulated, and the proposed methodology to address them is both well-founded and convincingly supported by experimental results.

**Essential References Not Discussed:**

There are no comments on this matter.

**Experimental Designs Or Analyses:**

The necessary experiments were well-designed, and the results were appropriately analyzed.

**Methods And Evaluation Criteria:**

The methodology clearly outlines how it aims to address the problem, and it is easy to understand how the proposed approach tackles the observed issues. Additionally, the evaluation was conducted using fair benchmarks in comparison with existing methods.

**Other Comments Or Suggestions:**

At line 630 in Appendix A, are you referring to Decoder instead of Encoder?

**Other Strengths And Weaknesses:**

The paper addresses a clearly defined and significant problem, with observations that are straightforward. The proposed methodology is well explained, making it easy to understand how it tackles the problem. Moreover, the work is supported by well-designed experiments and convincing results, and the analysis effectively addresses any potential concerns.

However, a few clarifications were needed, and the details are provided in the "Questions For Authors" section.

**Questions For Authors:**

1. In Figure 4(a), is the token identical to x₀ from the prefix encoder input understood from a teacher forcing perspective?
2. In the original Transformer decoder, self-attention and cross-attention are both present. In TracFormer, is it correct to understand that the decoder combines the self-attention causal mask and cross-attention into a single cross-attention operation? Was this design choice naturally adopted from a modeling perspective, or did it stem from a separate insight by the authors?
3. On line 242, the paper states, “Attention masks are used in cross-attention layers to ensure each decoder embedding only depends on desired input variables.” However, since equation (5) appears to merely apply a stride, I do not fully understand how it guarantees that the variable scope does not include X_C. Could you clarify this?
4. The proposed method employs two encoders (a prefix encoder and a suffix encoder). What is the fundamental advantage of using these two uni-directional encoders over a bidirectional encoder? Is this choice made solely for modeling purposes, or does it reflect the authors’ specific intuition?
5. In Figure 5, if the mask ratio is 1.0, does this correspond to the unconditional perplexity? If so, should we interpret that unconditional perplexity is worse than conditional perplexity?

**Relation To Broader Scientific Literature:**

Recently, there has been a trend in LLMs towards using simple transformers with tokenization, which has led to a significant reduction in architecture search even at smaller scales. In this regard, I find the scalable proposal in the paper to be impactful, although scaling in practice presents its own challenges.

**Theoretical Claims:**

I have reviewed Appendix G and did not identify any major issues.

---

> ### Author Rebuttal · Authors · 2025-03-31
>
> We thank the reviewer for their constructive feedback and for acknowledging that the paper addresses a clearly defined and significant problem.
>
> > At line 630 in Appendix A, are you referring to Decoder instead of Encoder?
>
> We thank the reviewer for pointing out this typo, and the paragraph title should be “Sparse Attention Masks of the Decoder”. We will fix the typo in the next version of the paper.
>
> > In Figure 4(a), is the token identical to x₀ from the prefix encoder input understood from a teacher forcing perspective?
>
> Yes, the target tokens are identical to the input. However, as described in Section 4.3 (Equations (5) and (6)), we set the cross-attention masks from the decoder to the encoder such that the decoder will not receive information about $x_0$ from the prefix encoder when predicting $X_0$. This is done by not attending to any token in the prefix encoder.
>
> > In TracFormer, is it correct to understand that the decoder combines the self-attention causal mask and cross-attention into a single cross-attention operation?
>
> There is no self-attention in the decoder of Tracformers, and the model predicts a token only using information from the encoder(s) through the cross-attention operations. This design choice (not including self-attention layers in the decoder) was made mainly for (inference-time) efficiency considerations, as the decoder can be fully parallelized when predicting multiple tokens.
>
> > However, since equation (5) appears to merely apply a stride, I do not fully understand how it guarantees that the variable scope does not include X_C. Could you clarify this?
>
> As shown in Figure 4, in the case of arbitrary-context (AC) generation, the masked tokens (variables not in $X_C$) are replaced by the mask token (<MASK>) so the model will not observe these tokens. In the case of contextual AR (CAR) generation, the condition $t’ < t$ in Equation (5) is used to guarantee that the model only observes previous tokens from the prefix encoder. In the next version of the paper, we will add more discussion to clarify how we guarantee the model does not observe tokens outside of $X_C$.
>
> Additionally, we agree with the reviewer that there are other ways to design the decoder masks to ensure the model does not observe variables that it should not during training. We plan to explore other design choices in future work.
>
> > The proposed method employs two encoders (a prefix encoder and a suffix encoder). What is the fundamental advantage of using these two uni-directional encoders over a bidirectional encoder?
>
> This design choice is made mainly because it aligns well with the contextual AR (CAR) generation paradigm, where the prefix encoder always encodes the full prefix context while the suffix encoder represents the given suffix tokens. It would be very interesting to see which (two unidirectional encoders versus one bidirectional encoder) performs better in the arbitrary-context (AC) generation paradigm, and we plan to explore it in future work. We will discuss this in the conclusion section in the next version of the paper.
>
> > In Figure 5, if the mask ratio is 1.0, does this correspond to the unconditional perplexity? If so, should we interpret that unconditional perplexity is worse than conditional perplexity?
>
> In Figure 5, we only vary the mask ratio from 0.1 to 0.9 in increments of 0.1. Therefore, the right-most conditional perplexity values correspond to a mask ratio of 0.9. In general, as the mask ratio increases, the conditional perplexity worsens because the model has access to less contextual information.

---

### Official Review · Reviewer_KPNd · 2025-03-14

**Overall Recommendation:** 4

**Summary:**

This paper proposes a novel model architectural modification of Transformers and demonstrate that the proposed model outperforms baselines on non-autoregressive (NAR) conditional generation tasks, especially when the mask pattern at inference is different from the mask pattern at training.

**Claims And Evidence:**

Most claims are supported by clear and convincing evidence.

One somewhat inaccurate claim is that at the end of Section 6.2, below Table 4, the authors claim that “Tracformer remains highly competitive, achieving results comparable to or better than larger models.”

But the *unconditional* generation perplexity (Table 4) of the proposed model does not compare favorably with strong baselines such as GPT-2, SEDD, and MDLM.

I understand that the proposed model is smaller, but more experiments (e.g. slightly scaling it up to match baseline model sizes) may be needed to more convincingly demonstrate its capability of unconditional generation.

This limitation does not undermine the authors claim regarding *conditional* generation tasks.

**Essential References Not Discussed:**

References are sufficiently discussed.

**Experimental Designs Or Analyses:**

I checked all experimental designs and analyses in the main paper. They are sound and valid.

**Methods And Evaluation Criteria:**

The proposed methods and evaluation criteria make sense.

**Other Comments Or Suggestions:**

N/A

**Other Strengths And Weaknesses:**

Strengths

1. originality: the multi-scale attention applied to the NAR context for better generalizability is novel. (Similarly designed sparse attention may have been used for AR Transformers to improve efficiency in long-context settings.)

2. clarity: the techniques and the results are clearly described.

Weaknesses

1. significance: the tasks, datasets, and evaluation metrics used in this paper mostly correspond to generic language modeling, emphasizing on fluency.
- This paper does not demonstrate that the proposed model outperforms baselines under some task-specific metrics (e.g. BLEU for translation, ROUGE for summarization, accuracy for reasoning, etc).
- These specialized tasks are commonly benchmarked for non-autoregressive language modeling papers. They provide important additional information since in these tasks, fluency is not the only important capability.

**Questions For Authors:**

1. In Figure 2:
- The pre-trained SEDD was only able to handle sequences of length 1024. How did you calculate the log likelihood of shorter sequences?
- Does SEDD output the exact model-predicted likelihood, or its lower bound (ELBO)?

2. In Table 1: What does a mask range of [0.25, 0.75] mean? Does it mean that out of the context window of 1024, the positions from 256 to 768 are masked, while other positions are visible?

3. I would like to learn from the authors about their insights on my feedback shared in the above sections of this review form.

**Relation To Broader Scientific Literature:**

This paper contributes to improving the generalizability of non-autoregressive (NAR) conditional language generation tasks, and compares favorably against SOTA discrete diffusion language models under OOD masking patterns.

**Theoretical Claims:**

This paper does not claim to include theory as its primary contributions.

---

> ### Author Rebuttal · Authors · 2025-03-31
>
> We thank the reviewer for their constructive feedback and for recognizing the novelty and clarity of our work.
>
> > I understand that the proposed model is smaller, but more experiments (e.g. slightly scaling it up to match baseline model sizes) may be needed to more convincingly demonstrate its capability of unconditional generation.
>
> We thank the reviewer for the suggestion. As pointed out by the reviewer, we would like to highlight that although the unconditional generation performance is less favored compared to the baselines, our model achieves significantly better conditional generation performance. We will adjust the descriptions and claims about the unconditional generation performance of Tracformers versus the baseline models.
>
> We agree with the reviewer that it would be valuable to conduct further experiments by scaling up Tracformer to assess how its performance improves with model size. Additionally, as discussed in the conclusion, another promising direction is exploring how Tracformer can be combined with recent advancements in training objectives for discrete diffusion models, such as diffusion language models. We plan to investigate these directions in future work.
>
> > the tasks, datasets, and evaluation metrics used in this paper mostly correspond to generic language modeling, emphasizing on fluency.
>
> We thank the reviewer for the suggestion on extending the evaluation metrics. We focus mainly on metrics that reflect generative modeling performance (conditional log-likelihood/perplexity) and fluency (MAUVE, BERT score) to study the fundamental problem of NAR models having bad conditional generation performance. Although fluency alone may not be sufficient to fully characterize generation quality, it provides insights into how well the generated text aligns with natural language distributions. Perplexity, on the other hand, serves as a general measure of the model’s generative modeling performance.
>
> To comprehensively evaluate Tracformers, we evaluated the BLEU-4 scores of Tracformer and the baseline model BART on the WikiText-103 dataset. The model was evaluated across six different masks. The evaluation setup follows Table 1. As indicated by the results, Tracformer outperforms BART consistently. We will include the results in the next version of the paper.
>
> | Mask ranges                     | Tracformer | BART  |
> | ------------------------------- | ---------- | ----- |
> | [0.25,0.75]                     | **0.524**  | 0.513 |
> | [0.5,1.0]                       | **0.540**  | 0.519 |
> | [0.1,0.4] & [0.6,0.9]           | **0.419**  | 0.405 |
> | [0,0.4] & [0.5,0.8]             | **0.339**  | 0.325 |
> | [0,0.25] & [0.75,1.0]           | **0.536**  | 0.523 |
> | [0,0.1] & [0.2,0.5] & [0.7,1.0] | **0.337**  | 0.322 |
>
> > The pre-trained SEDD was only able to handle sequences of length 1024. How did you calculate the log likelihood of shorter sequences?
>
> While SEDD was trained with sequences of length 1024, as described in the original paper (e.g., in their Section 5.3.2), they can handle sequences of small length. Specifically, when generating a sequence of length N, we only provide the first N tokens to the Transformer model. The authors of SEDD provided code to handle shorter sequences in their official GitHub repository, and we used their implementation. Note that all other models were also trained with sequences of length 1024, so the evaluation protocol is consistent across all models.
>
> > Does SEDD output the exact model-predicted likelihood, or its lower bound (ELBO)?
>
> As described in their paper, SEDD, like other discrete diffusion models, can only compute the Evidence Lower Bound (ELBO) rather than the exact model-predicted likelihood. This is a common limitation in diffusion models since computing the exact likelihood is intractable.
>
> > In Table 1: What does a mask range of [0.25, 0.75] mean? Does it mean that out of the context window of 1024, the positions from 256 to 768 are masked, while other positions are visible?
>
> Yes, the range of [0.25, 0.75] specifies the proportion of the context window that is masked. In the case of a context window of 1024 tokens, this means the positions from 256 to 768 are masked.
>
> > I would like to learn from the authors about their insights on my feedback shared in the above sections of this review form.
>
> We thank the reviewer for their valuable comments, which we found very helpful in improving the quality of our paper. For example, as suggested by the reviewer, we added new metrics to further strengthen the experiment section. We also fully agree that text generation quality can be assessed from multiple perspectives.

---

> > ### Comment · Reviewer_KPNd · 2025-04-01
> >
> > Thank you for the rebuttal!
> >
> > Re: “SEDD, like other discrete diffusion models, can only compute the Evidence Lower Bound (ELBO) rather than the exact model-predicted likelihood.”
> >
> > In light of that, some evaluation numbers might be slightly misleading: for example, Figure 2 claims to use the log-likelihoods predicted by SEDD, but as the authors stated in the rebuttal, SEDD can only compute the ELBO. Does it mean that the numbers reported in Figure 2 are in fact the ELBO numbers, instead of the actual model-predicted log likelihood? Furthermore, just because two query orders lead to different ELBOs does not necessarily mean that these two query orders will cause the model to predict different likelihoods. Of course, for SEDD, the exact likelihood is unavailable, but I think the authors should clearly note  (in applicable parts of the paper writeup) the conceptual gap between using a lower bound (ELBO) and using the exact model-predicted likelihood in their reported numbers.
> >
> > The proposed updates make sense. I encourage the authors to incorporate those updates in the paper draft.
> >
> > I think my rating of “4: Accept” remains accurate.

---

> > > ### Author Response · Authors · 2025-04-02
> > >
> > > We thank the reviewer for their additional feedback.
> > >
> > > > Does it mean that the numbers reported in Figure 2 are in fact the ELBO numbers, instead of the actual model-predicted log likelihood?
> > >
> > > The numbers in Figure 2 represent log-likelihoods but under constrained ordering schemes for token generation. For example, in Figure 2(a), we evaluate likelihoods using two ordering schemes—forward and reverse. We agree with the reviewer that these likelihoods do not directly represent the overall likelihood or ELBO of SEDD, as computing the ELBO requires sampling or enumerating various unmasking strategies and weighing the resulting likelihoods for each strategy.
> > >
> > > The illustrative experiment is not intended to analyze the ELBO or the overall likelihood of the SEDD model but rather to demonstrate how generative performance is influenced by the mask/unmask strategy. This further leads to the “query generalization” problem discussed in the last two paragraphs of Section 3.
> > >
> > > We will include the above discussion and note the conceptual gap between ELBO and likelihood in the next version of the paper.
> > >
> > > > I think the authors should clearly note (in applicable parts of the paper writeup) the conceptual gap between using a lower bound (ELBO) and using the exact model-predicted likelihood in their reported numbers
> > >
> > > In the next version of the paper, we will clarify that we only report ELBO numbers instead of log-likelihoods for discrete diffusion models such as SEDD and MDLM since computing likelihood is intractable. We will also incorporate other suggestions into the next version of the paper, as detailed in the rebuttal.

---

### Official Review · Reviewer_sirR · 2025-03-15

**Overall Recommendation:** 4

**Summary:**

This paper is motivated by the fact that Non Auto-regressive (NAR) generative models do not work very well for conditional generation. The paper proposes Tracformers, a transformer-based architecture robust for conditional generation in the more difficult NAR setting: this is done through using *multiple context levels*, in order to force the model being able to use only local subset of the context when training for conditional generation, which authors call *query generalization*. After describing their proposed architecture, they describe two settings, Contextual AR generation and Arbitrary Context (AC) generation, for which they describe the instantiation of the model and the loss. Experiments compare, in both settings, a model built with Tracformers to basic (BERT and BART) and a scaled-up version to recent (diffusion-based) generative NAR models, using conditional complexity on Wiki103, 1B and Lambada.

**Claims And Evidence:**

The paper claims that *Tracformer’s multi-scope attention mechanism and specialized encoder-decoder design enable robust conditional generation performance*, which is verified by the experiments. However, as pointed out in the conclusion, the larger model proposed is the size of a GPT-2 model, and the question remains if the reduction of interactions in self-attention in the Tracformer layer will significantly restrict the model's expressiveness at larger scales (and in particular, for larger encoded sequences).

**Essential References Not Discussed:**

I have no knowledge of any essential reference missing.

**Experimental Designs Or Analyses:**

- The experiments compare the Tracformer with the main types of models corresponding to AR and CAR settings; and then, to state-of-the-art NAR generative models.
- Models are compared across various mask ratios.
- The analysis is extensive, and further ablation studies are provided in the supplementary material.

**Methods And Evaluation Criteria:**

The method relies on an encoder-decoder architecture, where:
- The encoder uses increasing context length with the layers (exponential growth), with a maximal number of attended positions
- The decoder work through cross-attention only, and making independent predictions at different positions; attending first encoded representations built on large scope and moving towards local information.
The description of the architecture relies on mask to restrict attention to the desired scope.

Depending on the setting (CAR implies auto-regressive generation with partial knowledge of the future, while AC is completely NAR) and the available input context, different mask schemes and losses are implemented.

Evaluation is made mainly on conditional perplexity of the generated sequences, as well as infilling performance with MAUVE and Bert score.

**Other Comments Or Suggestions:**

- The abstract is quite unclear in retrospect; terms that are well defined later (conditional probability query) appear obscure.
- L238: while this remark may only be here to provide intuition for the design of the model, it would be better if it was backed by a reference.
- Footnote 1 is unclear: I don't see how it matches Equation 3.

**Other Strengths And Weaknesses:**

- This paper propose a promising idea which should be further explored.
- While this paper is generally well written, Section 4 stays difficult to follow.

**Questions For Authors:**

I have no question for the authors.

**Relation To Broader Scientific Literature:**

This paper's contribution is a new architecture for NAR modeling, and could be used with dedicated NAR models, such as discrete diffusion models.

**Theoretical Claims:**

No theoretical claims are made in the paper.

---

> ### Author Rebuttal · Authors · 2025-03-31
>
> We thank the reviewer for their constructive feedback and for recognizing Tracformers as a promising architecture for addressing the challenges NAR models face in conditional generation.
>
> > The abstract is quite unclear in retrospect; terms that are well defined later (conditional probability query) appear obscure.
>
> We appreciate the reviewer’s suggestion. In the next version of the paper, we will clarify the term “conditional probability query” directly in the abstract, referring to the set of tokens/variables provided during NAR generation.
>
> > L238: while this remark may only be here to provide intuition for the design of the model, it would be better if it was backed by a reference.
>
> While the main goal of this sentence is to give intuition, it can indeed be supported by references such as [1], which highlights the effectiveness of a top-down hierarchical scheme in long text generation. We will add the reference and further discuss it in the next version of the paper.
>
> > Footnote 1 is unclear: I don't see how it matches Equation 3.
>
> We thank the reviewer for pointing out the typo in Footnote 1. According to the definition in Equation 3, we have that $\phi^{l-1}_{t-2^{l-1}}$ = {$t’ : t’ \geq 1, 0 \leq t - 2^{l-1} - t’ < 2^{l-1}$}
>
> and
>
> $\phi^{l-1}_{t}$ = {$t’ : t’ \geq 1, 0 \leq t - t’ < 2^{l-1}$}.
>
> Taking the union of the two sets leads to
>
> $\phi^{l}_{t}$ = {$\{t’ : t’ \geq 1, 0 \leq t - t’ < 2^{l}\}$}.
>
> We will fix it in the next version of the paper.
>
> > While this paper is generally well written, Section 4 stays difficult to follow.
>
> In our humble opinion, the main reason Section 4 is more challenging to follow than the other sections is the indexing of the variable scopes for different tokens, which is necessary to rigorously define Tracformer. To improve clarity, we will add a figure with one example Tracformer layer where each feature embedding is labeled with the corresponding variable scope. This figure will then be frequently referenced throughout the model description.
>
> [1] Wan, Kaiyang, Honglin Mu, Rui Hao, Haoran Luo, Tianle Gu, and Xiuying Chen. "A Cognitive Writing Perspective for Constrained Long-Form Text Generation." arXiv preprint arXiv:2502.12568 (2025).

---

### Decision · Program_Chairs · 2025-05-01

**Decision:**

Accept (poster)

**Comment:**

The paper addresses conditional generation in NAR, and proposes a model called Tractable Transformers (TracFormer). Essentially, the authors devised a multi-scope attention that uses only local subset of the context.

While reviewers believe the idea is interesting, there are a few drawbacks of the experiments. For example, the language models used are generally small, and the evaluation is mostly PPL oriented, except for the text infilling experiment which uses MAUVE and BERT scores (Table 1).